# Carbon emissions from the 2023 Canadian wildfires

Brendan Byrne[1✉], Junjie Liu[1,2], Kevin W. Bowman[1,3], Madeleine Pascolini-Campbell[1], Abhishek Chatterjee[1], Sudhanshu Pandey[1], Kazuyuki Miyazaki[1], Guido R. van der Werf[4], Debra Wunch[5], Paul O. Wennberg[2,6], Coleen M. Roehl[2] & Saptarshi Sinha[7]

The 2023 Canadian forest fires have been extreme in scale and intensity with more than seven times the average annual area burned compared to the previous four decades[1]. Here, we quantify the carbon emissions from these fires from May to September 2023 on the basis of inverse modelling of satellite carbon monoxide observations. We find that the magnitude of the carbon emissions is 647 TgC (570–727 TgC), comparable to the annual fossil fuel emissions of large nations, with only India, China and the USA releasing more carbon per year[2]. We find that widespread hot–dry weather was a principal driver of fire spread, with 2023 being the warmest and driest year since at least 1980[3]. Although temperatures were extreme relative to the historical record, climate projections indicate that these temperatures are likely to be typical during the 2050s, even under a moderate climate mitigation scenario (shared socioeconomic pathway, SSP 2–4.5)[4]. Such conditions are likely to drive increased fire activity and suppress carbon uptake by Canadian forests, adding to concerns about the long-term durability of these forests as a carbon sink[5–8].

Canadian forests cover a vast area of nearly 362 million ha (ref. 9), amounting to 8.5% of the global forested area[10]. These forests are an important sink of carbon, absorbing fossil carbon dioxide ($CO_2$) from the atmosphere and slowing the pace of climate warming[11,12]. However, climate change is increasing forest fire activity, acting to suppress the carbon uptake capacity of these forests[13]. Although more frequent fires have been widespread, 2023 has seen forest fires on an extreme scale. With 15 million ha of Canadian forests burned (about 4% of forest area)[1], 2023 saw more than seven times (8 $\sigma$) the average burned area over the preceding 40 years (1983–2022 mean, 2.2 million ha; range, 0.2–7.1 million ha)[1]. The adverse societal impacts of these fires are clear: 232,000 evacuations and poor air quality affecting millions[14]. However, the carbon emissions from the fire events remain uncertain. In this study, we quantify these emissions through inverse modelling of satellite observations of carbon monoxide (CO). Then, we examine concurrent climate anomalies and projected changes in the prevalence of hot–dry weather under climate change. Finally, we discuss the implications of fires for the Canadian carbon budget.

## Fire emissions

Fire carbon emissions can be tracked from space using bottom-up and top-down approaches. Bottom-up approaches use satellite observations to track fire activity, such as burned area[15] or fire radiative power[16]. Emissions of $CO_2$, CO and other trace gases are then estimated by combining the estimates of fire activity with quantities such as fuel loads and emission factors. Although these bottom-up estimates are continually improving, inventories can vary significantly

in global and regional trace gas and aerosol emission estimates[15,17]. Top-down approaches provide a method for refining bottom-up trace gas emission estimates by optimally scaling emission estimates to be consistent with the observed concentrations of trace gases in fire plumes. A strength of this approach is that it integrates emissions from both flaming and smouldering combustion to capture net emissions.

In this study, we perform top-down estimates of CO emissions from the 2023 Canadian fires based on observational constraints from the TROPOspheric monitoring instrument (TROPOMI) space-based CO retrievals (Fig. 1a,b). These estimates are performed using three different bottom-up fire emission inventories: the global fire emissions database (GFED4.1s)[15], the global fire assimilation system v.1.2 (GFAS)[16] and the quick fire emissions dataset v.2.6r1 (QFED)[18]. For each inversion, the combined carbon emissions released as CO and $CO_2$ ($CO_2$ + CO) are then estimated using the $CO_2$/CO emission factors from the same bottom-up database. The $CO_2$/CO emission ratios can be highly variable, adding uncertainty to our analysis. We incorporate some of this uncertainty here as each bottom-up database has different mean emission ratios for Canadian forests (range, 7.7–10.8 gC of $CO_2$ per gC of $CO_2$). Details for these inversions are provided in the methods and a description of the inversion results and evaluation of the performance of the top-down estimates are provided in Supplementary Information sections 1 and 2). We find the top-down estimates are relatively insensitive to choices about inversion configuration but do show sensitivity to prescribed hydroxyl radical (OH) abundances[19], which determine the atmospheric lifetime of the CO emitted (Supplementary Information section 1 and Supplementary Fig. 1).

[1]Jet Propulsion Laboratory, California Institute of Technology, Pasadena, CA, USA. [2]Division of Geological and Planetary Sciences, California Institute of Technology, Pasadena, CA, USA. [3]Joint Institute for Regional Earth System Science and Engineering, University of California, Los Angeles, CA, USA. [4]Meteorology & Air Quality Group, Wageningen University and Research, Wageningen, The Netherlands. [5]Department of Physics, University of Toronto, Toronto, Ontario, Canada. [6]Division of Engineering and Applied Science, California Institute of Technology, Pasadena, CA, USA. [7]Department of Energy, Environmental, and Chemical Engineering, Washington University in St. Louis, St. Louis, MO, USA. ✉e-mail: brendan.k.byrne@jpl.nasa.gov

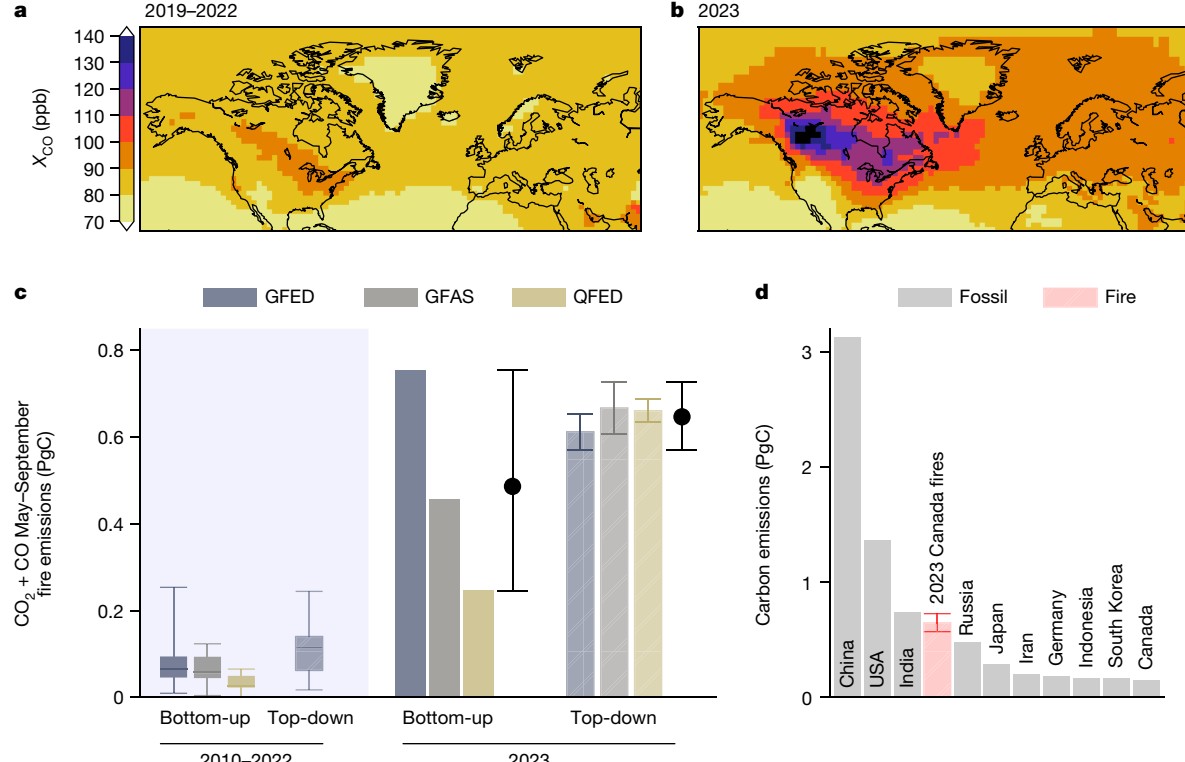

**Fig. 1 | CO enhancements and fire emission estimates. a–c**, May–September TROPOMI dry-air mole fractions of CO ($X_{CO}$) averaged over 2019–2022 (**a**) and for 2023 (**b**) aggregated to a 2° × 2.5° grid. **c**, Canadian forest fire carbon emissions (from CO and $CO_2$) for the 2023 May–September fire season, compared with fire emissions during 2010–2022 (distribution shown by box-and-whisker plots). Top-down emissions over 2010–2022 are estimated from MOPITT (2010–2021) and TROPOMI (2019–2022) CO retrievals. **d**, A comparison of May–September Canadian fire emissions with 2022 territorial fossil carbon emissions for the ten largest emitting countries, obtained from Global Carbon Budget 2022[2].

Figure 1c shows the bottom-up and top-down $CO_2$ + CO carbon emissions from fires during May–September 2023. The bottom-up datasets show large differences, ranging from 234 to 735 TgC (mean of 469 TgC). This range is reduced by 69% in the top-down estimates (570–727 TgC), which also give a larger mean estimate of 647 TgC. Emissions during 2023 far exceed typical Canadian forest fire emissions, with 2010–2022 average emissions of 29–82 TgC for the bottom-up inventories and 121 TgC for top-down estimates (Supplementary Fig. 2). To contextualize these numbers, we compare the top-down estimates to annual national fossil fuel emissions for the ten largest emitters (Fig. 1d). The 5 month 2023 emissions are more than four times larger than Canadian annual fossil fuel emissions (149 TgC yr$^{-1}$) and comparable to India's annual emissions (740 TgC yr$^{-1}$).

Fire activity is affected by several complex drivers, including fuel traits[20] and ignition probability[21]. However, fire weather—hot and dry conditions—has been shown to be extremely important in driving fire behaviour[22]. Climate data show an exceptionally hot and dry fire season for Canadian forests during 2023 (Fig. 2). This was the driest January–September period for Canadian forests since at least 1980, with about 86% of forested area having below-average precipitation and about 52% being more than 1 s.d. below the 2003–2022 average (Supplementary Fig. 4). May–September 2023 was the warmest since at least 1980, with about 100% of the forest area above average and about 90% being more than 1 s.d. above the 2003–2022 average. Similarly, the vapour pressure deficit (VPD), which is closely associated with fire activity[22–24], was the third highest since 1980, including 85% of the forest area being above average and about 54% being more than 1 s.d. above the 2003–2022 average.

Although hot–dry conditions were widespread across Canadian forests, there are two notable regional patterns. Western Quebec (49°–55° N, 72°–80° W), which is typically relatively wet (Supplementary Fig. 5a), had exceptionally dry conditions during 2023, with precipitation through September being 23.7 cm (49%) below average. Coupled with extreme heat and VPD during June–July, fire emissions in this region contributed about 15% of the national total (Supplementary Fig. 6). The other notable region was northwestern Canada near the Great Slave Lake (57°–62° N, 110°–125° W). This region is drier than western Quebec on average, with about half the annual precipitation. However, 2023 was exceptional, with both a large precipitation deficit of 8.1 cm (27% of January–September total) and exceptionally warm conditions throughout May–September (+2.6 °C) (Supplementary Fig. 6). This region contributed about 61% of the total Canadian forest fire emissions.

## Fires and climate

The relationship between climate variability and fire emissions for Canadian forests is examined in Fig. 3, which shows fire emissions as a function of temperature and precipitation $Z$-scores over 2003–2023 for the 0.5° × 0.625° grid cells, in which $Z$-scores are the anomalies divided by the standard deviation. May–September emissions are lowest for combined cool–wet conditions (5.2 gC m$^{-2}$), whereas emissions increase when either temperature is above average (19.5 gC m$^{-2}$) or precipitation is below average (9.2 gC m$^{-2}$). However, emissions are largest for combined warm–dry conditions (35.7 gC m$^{-2}$). In particular, fire emissions are much increased during exceptionally hot and dry conditions (99.6 gC m$^{-2}$, temperature $Z > 1$ and precipitation $Z < -1$). These hot–dry conditions were much more prevalent in 2023 than in preceding years, with a mean May–September T2M $Z$-score of 2.3 and a precipitation $Z$-score of −1.1 across grid cells, explaining why fire emissions were extreme during 2023. Notably, the number of individual fires during 2023 was not unusual, with 6,623 relative to a 10 yr average of

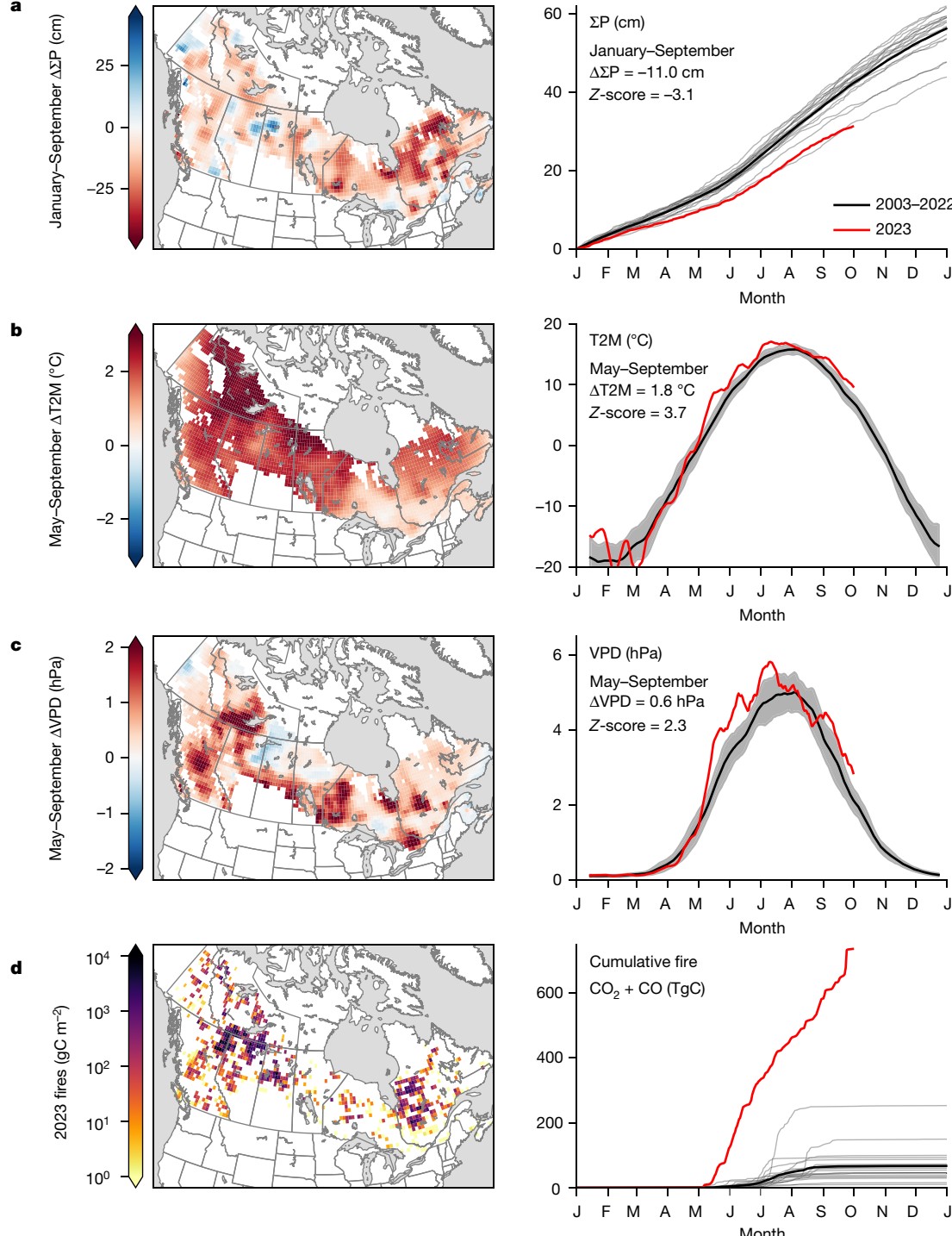

**Fig. 2 | Canadian forests climate anomalies for 2023 relative to 2003–2022 mean. a–d**, Maps (left) and time series (right) of CPC global unified gauge-based cumulative precipitation ($\sum P$) (**a**), MERRA-2 2 m temperature (with 2 week running mean) (**b**), MERRA-2 VPD (with 2 week running mean) (**c**) and fire $CO_2 + CO$ emissions from the GFED4.1s database (**d**). All maps are shown at a spatial resolution of $0.5° \times 0.625°$ and Z-scores are for the area-mean of Canadian forests. Note that GFED4.1s is shown instead of the inversion results because those are at a coarser spatial resolution and cover a shorter time period, maps of prior and posterior mean fire emissions are shown in Supplementary Fig. 14. Months are shown from January (J) to December (D).

5,597 (ref. 25). Yet, probably primarily driven by these hot–dry conditions[24], many of these fires grew to enormous sizes with hundreds of megafires (greater than 10,000 ha) recorded.

Next, we examine future climate conditions in the region and how they compare to the concurrent climate conditions that led to the massive fires. Figure 3 shows the decadal mean temperature and precipitation Z-scores for the median of 27 models from the coupled model intercomparison project phase 6 (CMIP6)[26] under the moderate-warming shared socioeconomic pathway (SSP) 2–4.5 (ref. 4). Large projected temperature increases are found to occur, with average temperatures in the 2050s similar to 2023. More modest increases in precipitation are projected, indicating a 'speeding up' of the water cycle, in which both evaporation and precipitation rates increase (Supplementary Fig. 12 shows ensemble distribution). Studies indicate that

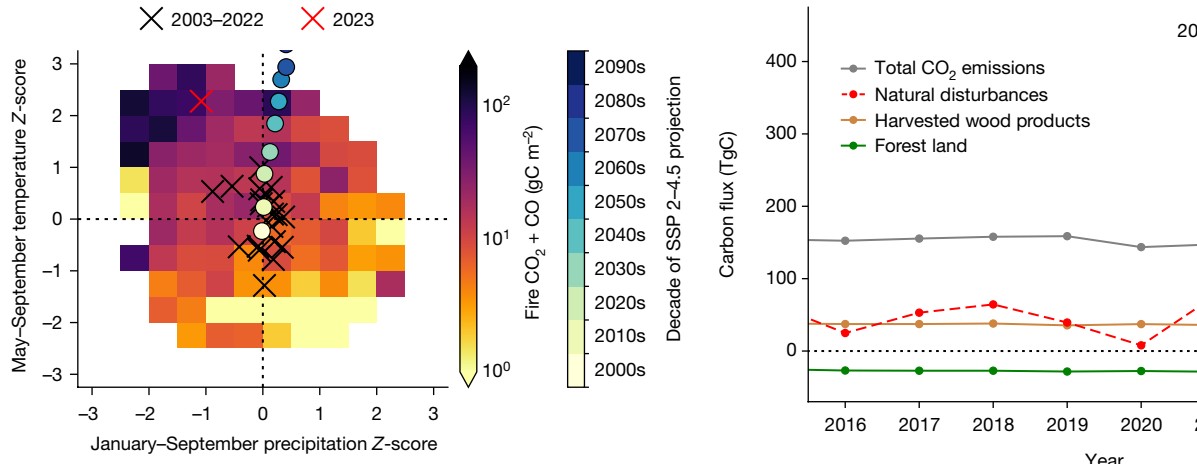

**Fig. 3 | Relationship between fire emissions and climate anomalies.**
Mean May–September GFED4.1s $CO_2$ + CO fire emissions as a function of May–September T2M Z-score and January–September precipitation Z-score for each $0.5° \times 0.625°$ forested grid cell during 2003–2023 (using 2003–2022 as a baseline). For individual years, the mean Z-scores across forested grid cells are shown with 'X'. The projected decadal-mean temperature and precipitation Z-scores for the median CMIP6 model under SSP 2–4.5 are shown by the circles. The CMIP6 Z-scores are calculated using the 2000–2019 period as a baseline but use the reanalysis 2003–2022 standard deviations (see section on 'Climate data'). The historical and projected T2M and precipitation over the Canadian boreal forests simulated by the CMIP6 ensemble are shown in Supplementary Fig. 12.

**Fig. 4 | Canada's NGHGI $CO_2$ emissions and removals compared with the 2023 Canadian fires.** Lines show the annual net emissions or removals from managed forest land (green), harvested wood products (brown), natural disturbances that are not counted towards Canada's emissions (red) and the economy-wide net $CO_2$ emissions (grey). The top-down estimates of the 2023 $CO_2$ + CO fires emissions over managed land are shown in black. Total $CO_2$ emissions, harvested wood products and forest land emissions and removals were obtained from Table A11-1 of the NGHGI[32], whereas natural disturbances were obtained from Table 6-5 of the NGHGI. All quantities presented are in units of teragrams of carbon (1 TgC = 1 MtC = 1,012 gC), which can be converted to units of megatonnes of $CO_2$ (Mt$CO_2$) by multiplying by a factor of 3.664.

the combined effect will result in regional increases in moisture deficits for Canadian forests through the end of the twenty-first century[6,27,28]. Beyond the 2050s, average temperature and precipitation conditions are projected to exceed the historical range. These changes will impact the boreal carbon cycle in many ways, such as changing fuel loads and species composition, which complicates projections of future fire activity. However, increases in boreal fire emissions linked to warming have been reported over recent decades[13,27,29,30] and several studies have projected further increases in Canadian fire activity with future warming[5–8]. Thus, we find that warming, coupled with regionally increasing moisture deficits, is likely to drive increased fire carbon emissions from Canadian forests.

## Canadian carbon budget implications

As a party to the Paris Agreement, Canada is obligated to track economy-wide greenhouse gas (GHG) emissions and removals in a national GHG inventory (NGHGI). This includes tracking emissions and removals from 'managed' lands, for which human interventions and practices have been applied to perform production, ecological or social functions[31]. However, the 2006 Intergovernmental Panel on Climate Change (IPCC) guidelines for national GHG inventories[31] and Canadian NGHGI[32] differ in how emissions and removals over managed lands are categorized. The IPCC guidelines treat all emissions and removals on managed land as anthropogenic, whereas the Canadian NGHGI treats 'natural disturbances' as non-anthropogenic. This difference in categorization leads to large differences between the Canadian NGHGI and an estimate using the IPCC guideline definitions.

Figure 4 shows that NGHGI removals on managed forest land are almost exactly compensated by emissions from harvested wood products, such that the total $CO_2$ emissions for Canada are dominated by the energy sector (more than 90% of net emissions). However, we see that natural disturbances are shown to be of considerable magnitude, amounting to nearly 60% of total $CO_2$ emissions in 2021. The 2023 CO + $CO_2$ fire emissions across managed Canadian forests (see section

on 'Managed land') are estimated to be 421 (388–461) TgC, amounting to 2.5–3 years of economy-wide $CO_2$ emissions.

Regardless of their characterization, fire carbon emissions will affect the growth rate atmospheric $CO_2$. As such, monitoring changes in the carbon budget across both managed and unmanaged land is important. Including all land in the Canadian carbon budget, top-down estimates find that Canadian ecosystems are a sink of $CO_2$ when constrained by either in situ or space-based $CO_2$ observations. Using both data types, an ensemble of atmospheric $CO_2$ inversion systems report that Canadian carbon stocks increased 366 ± 88.6 TgC yr$^{-1}$ over 2015–2020[11], contributing about 30% of the net land carbon sink. Similarly, space-based biomass estimates find carbon accumulation in Canadian boreal forests, although smaller in magnitude[13,33,34]. Thus, Canadian forests play an important role in mitigating anthropogenic emissions, slowing the rise of atmospheric $CO_2$. The large carbon release resulting from the 2023 Canadian fires puts into question the durability of this sink. Others[13] showed that fires have acted to suppress the carbon uptake potential of Canadian forests over the past 30 years. Although Canadian forests have historically experienced large stand-replacing fires at infrequent intervals of 30 to more than 100 years[35–37], increases in fire frequency will probably reduce biomass recovery and affect species composition[37–40]. It has also been argued that fire, insects and droughts may already be driving Canadian forests into a carbon source[41,42]. In the extreme case that expansive fires, such as that of 2023, become the norm (burning 4% of Canadian forest area), all Canadian forests could burn every 25 years. So, although the magnitude is uncertain, it is likely that increasing fire activity in Canadian forests will reduce the capacity of these Canadian forests to continue to act as a carbon sink.

The role of Canada's fire management strategy in managing fire carbon emissions also deserves some discussion. Fire management strategies require balancing several considerations, including socio-economic costs, ecological impacts and carbon emissions. Canada's present strategy adopts a risk-based approach, for which decisions on whether or not to suppress fires are made on a fire-by-fire basis[43], with

differing priorities across provinces and territories. Understanding how fire regimes will change with climate change is thus of high importance, for future decision criteria and costing.

## Conclusions

The 2023 fire season was the warmest and driest for Canadian forests since at least 1980, resulting in vast carbon emissions from forest fires. Using TROPOMI CO retrievals, we estimate the total May–September $CO_2$ + CO emissions from these fires to be 647 TgC (range 570–727 TgC), comparable in magnitude to India's annual fossil fuel $CO_2$ emissions. The 2023 warmth was exceptional based on the last 44 years but CMIP6 climate models project that the temperatures of 2023 will become normal by the 2050s. Such changes are likely to increase fire activity[5–8], risking the carbon uptake potential of Canadian forests. This will impact allowable emissions for reaching warming targets, as reduced carbon sequestration by ecosystems must be compensated for by adjusting anthropogenic emissions reductions.

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

# Article

## Methods

### Climate data

Precipitation estimates were derived from Climate Prediction Center (CPC) global unified gauge-based analysis of daily precipitation data provided by the National Oceanic and Atmospheric Administration from their website at https://psl.noaa.gov (refs. 44,45). MERRA-2 2 m temperature (T2M) and dew point temperature at 2 m (T2MDEW) were obtained from the single-level diagnostics file[3]. VPD was calculated from these quantities using:

$$VPD = es - ea,$$

where es is the saturation vapour pressure and ea is the vapour pressure, calculated from T2MDEW and T2M, respectively, using the formulation of ref. 46. The $Z$-scores for precipitation, T2M and VPD were calculated relative to the 20-year baseline of 2003–2022; for example, for T2M this is calculated as:

$$Z-\text{score}_{year} = \frac{T2M_{year} - \text{mean}(T2M_{2003-2022})}{\text{std}(T2M_{2003-2022})},$$

where $T2M_{year}$ is the May–September mean T2M for a given year and $T2M_{2003-2022}$ is the 20-element ensemble of May–September mean T2Ms during 2003–2022.

CMIP6 data were downloaded from the Canadian Climate Data and Scenarios website (https://climate-scenarios.canada.ca/?page=cmip6-scenarios). We examine the ensemble median of 27 models provided on a $1° \times 1°$ grid (technical documentation at https://climate-scenarios.canada.ca/?page=pred-cmip6-notes). The models included are based on data availability and are tabulated at https://climate-scenarios.canada.ca/?page=cmip6-model-list. T2M and precipitation are analysed for the historical and future scenarios. We combine the historical simulation with SSP 2–4.5 (ref. 4), shown in the main text or SSP 5–8.5 as shown in Supplementary Fig. 7.

The $Z$-scores are calculated from the median of the CMIP6 ensemble by calculating the temporal mean of the median model mean over 2000–2019 for a given grid cell, whereas the reanalysis data are used to estimate internal variability. Therefore, the T2M $Z$-score is calculated as:

$$Z-\text{score}_{year} = \frac{T2M_{\text{CMIP year}} - \text{mean}(T2M_{\text{CMIP2000-2019}})}{\text{std}(T2M_{\text{MERRA2003-2022}})}$$

### Sources and sinks

**Fossil CO emissions.** Anthropogenic CO emissions are obtained from the community emissions data system (CEDS) for historical emissions[47]; specifically we use version CEDS-2021-04-21 (ref. 48).

**Prior fire $CO_2$ and CO emissions.** Fire $CO_2$ and CO emissions are obtained from the GFED, GFAS and QFED databases. GFED4.1s[15] provides estimates of biomass burning using a biogeochemical model ingesting MODIS 500 m burned area[49] in combination with 1 km thermal anomalies and 500 m surface reflectance observations to estimate burned area associated with small fires using a statistical model[50]. These data were downloaded from https://www.globalfiredata.org/. GFAS v.1.2 provides estimates of daily biomass burning emissions by assimilating MODIS fire radiative power observations[16]. These data were downloaded from the atmosphere data store (https://ads.atmosphere.copernicus.eu). We use v.2.6 of the QFED gridded emission estimates[18]. These data were downloaded from https://portal.nccs.nasa.gov/datashare/iesa/aerosol/emissions/QFED/v2.6r1/0.25/QFED/. For all biomass burning datasets, we release fire emissions at the model surface but incorporate a 3 hourly diurnal cycle based on ref. 51. Year-specific emissions are used for the prior in the atmospheric CO inversions.

**Biogenic emissions, atmospheric CO production and OH data.** Biogenic emissions, atmospheric CO production and OH data were all derived from the outputs of the MOMO-Chem chemical data assimilation[52]. An updated version of the tropospheric chemistry reanalysis v.2 (TCR-2)[53] produced using MOMO-Chem is used to evaluate the atmospheric production and loss of CO. The reanalysis is produced through the assimilation of several satellite measurements of ozone, CO, $NO_2$, $HNO_3$ and $SO_2$. The chemical loss of CO was estimated using the reanalysis OH fields. Because of the multiconstituent data assimilation, the reanalysis OH shows improved agreements in global distributions over remote oceans in comparison with the ATom aircraft measurements from the surface to the upper troposphere[53]. Constraints obtained for OH profiles have a large potential to influence the chemistry of the entire troposphere, including oxidation of non-methane hydrocarbons (NMHCs) to estimate the chemical production of CO. The biogenic emissions at the surface were obtained from the model of emissions of gases and aerosols from Nature v.2.1 (MEGAN2.1)[54]. Year-specific fields were only available through 2018 and estimates for that year are repeated for more recent years. We also perform a supplemental sensitivity analysis for the impact of prescribed OH abundances on inferred emissions using the fields of ref. 55, which are commonly used for GEOS-Chem methane inversions[56].

### CO retrievals

**TROPOMI.** TROPOMI is a grating spectrometer aboard the ESA Sentinel-5 Precursor (S5P) satellite which measures Earth-reflected radiances[57]. CO total column densities are retrieved in the shortwave infrared (around 2.3 μm) using the shortwave infrared CO retrieval algorithm[58,59]. TROPOMI CO retrievals[60] were downloaded from the Copernicus data space ecosystem (https://dataspace.copernicus.eu/). We use S5P RPRO L2 CO (processor v.2.4.0) through 25 July 2022, then switch to S5P OFFL L2 CO for more recent data (processor v.2.5.0 or 2.5.0). Retrieved CO total column densities are then converted to dry-air mole fractions of CO ($X_{CO}$) using the dry-air surface pressure and hypsometric equation. The column averaging kernel is similarly converted to mole-fraction space. Individual retrievals (quality flag $\geq 0.5$) from each orbit are aggregated into super-observations using the model grid ($2° \times 2.5°$).

The retrieval uncertainty on super-observations is taken to be the mean uncertainty on all retrievals in a given super-observation. This approach is used because systematic errors may exist between retrievals, such that assuming random errors would underestimate the true retrieval error. For assimilation into NASA carbon monitoring system-flux (CMS-Flux), we calculate observational errors that incorporate error in the atmospheric transport model. For this, we follow the approach of ref. 61. First, we perform a forward model simulation with the prior fluxes for 2019–2023. Then we take the observational uncertainty to be the standard deviation between the simulated and real TROPOMI super-observations over a moving window of 30° latitude, 30° longitude and 30 days (across all years). The uncertainties estimated using this approach range over 3.5–14.3 ppb (5–95 percentiles), whereas retrieval errors range over 1.4–4.9 ppb. Thus, the observational errors are dominated by representativeness errors.

**MOPITT.** We use the MOPITT (measurements of pollution in the troposphere) satellite thermal-infrared–near-infrared (TIR–NIR) CO retrieval. Version 9 (L2V19.9.3)[62] is used from 2009 to 31 October 2022, whereas L2V19.10.3.beta is used from 1 November 2022 onwards. These data were downloaded from the EarthData ASDC (https://asdc.larc.nasa.gov/data/MOPITT/). As with TROPOMI, profile retrievals were converted into dry-air mole fractions of CO ($X_{CO}$) for assimilation; however, unlike TROPOMI, we do not generate super-observations but instead assimilate individual observations. This is because the footprint of MOPITT retrievals ($22 \times 22$ km$^2$) is much coarser than TROPOMI retrievals ($3 \times 7$ km$^2$).

**TCCON.** The total carbon column observing network (TCCON) consists of ground-based Fourier transform spectrometers which retrieve $X_{CO}$, $X_{CO2}$ and other species from observations of solar radiation[63]. In this study, we examine GGG2020 (ref. 64) TCCON data from Park Falls[65] and East Trout Lake[66]. These data were obtained from the TCCON Data Archive hosted by CaltechDATA at https://tccondata.org. Super-observations are created for each site as hourly averages; we only include hours with five or more observations.

### Atmospheric CO inversions

We perform a series of CO inversion analyses using the CMS-Flux atmospheric inversion system. This inversion model is descended from the GEOS-Chem adjoint model[67] and has been used for $CO_2$ (refs. 68,69) and CO inversion analyses[70]. The inversions in this study are all performed globally at 2° × 2.5° spatial resolution using MERRA-2 reanalysis. CEDS anthropogenic emissions, biogenic atmospheric CO production, direct biogenic CO emissions and fire emissions (from GFED4.1s, GFAS or QFED) and atmospheric OH fields are all prescribed in the forward simulations (see section on 'Sources and sinks'). Four-dimensional variational data assimilation (4D-Var) is used to optimize scaling factors on the net surface flux for each grid cell (combined anthropogenic, fire and direct biogenic CO flux). The posterior CO fluxes are then decomposed into anthropogenic, fire and biogenic fluxes using the fractional contribution of the prior (an approach widely used for CO inversions).

A series of MOPITT $X_{CO}$ inversions are performed over 2010–2021. Weekly fluxes are optimized over the period 7 November of the preceding year (YYYY − 1) to 1 February of the next year (YYYY + 1), the optimized fluxes in the desired year are retained (YYYY) and the fluxes outside this period are discarded as spin-up or spin-down. These inversions are performed using the GFED4.1s fire inventory. Prior uncertainties on emissions are assumed to be proportional to the emissions, with a scale factor uncertainty of 200%.

TROPOMI $X_{CO}$ inversions are performed over 2019–2023. These inversions are performed over a truncated period of 1 April to 30 September, with April then being discarded as spin-up. Several different inversion configurations are used to quantify the uncertainty in posterior fluxes due to both Bayesian posterior uncertainties and systematic choices about error specification and inversion configuration, both of which have been shown to contribute significantly to inversion error estimates[11].

Three ensembles of inversions are performed on the basis of the three different prior fire inventories: GFED4.1s, GFAS or QFED (Extended Data Fig. 1a). Each prior inventory was subjected to four different experimental configurations (Extended Data Fig. 1b). In one case, the $X_{CO}$ super-observations error is taken to be the mean retrieval uncertainty across all retrievals included in a given super-observation. This approach typically gives an uncertainty of 1.3–4.9 ppb. The other case uses an observational error estimate that incorporates representativeness errors (see section on 'TROPOMI'), which are typically between 3.5 and 14.3 ppb. The experimental configurations also differ by the treatment of prior uncertainties on the fluxes. These uncertainties are not well known a priori, thus we use two very different approaches. In the first approach, we assume that the errors on fluxes are equal to 200% of the prior flux estimate. In the second approach, we assume that flux uncertainties are near constant in flux units (scale factor uncertainty times control flux is constant, this is truncated to scale factors uncertainties between 0.25 and 1,000). Finally, we also vary the temporal optimization to either 3 or 7 days. As with the prior flux uncertainties, there are many possible choices for temporal optimization, so we choose two reasonable estimates to quantify the sensitivity to this choice. The spread in maximum a posteriori estimates across these different set-ups gives an indication of the uncertainty in estimated fluxes due to the set-up decisions.

We also estimate the Bayesian posterior uncertainty (Extended Data Fig. 1c), which derives from uncertainties in the prior fluxes and observations. This uncertainty is estimated using the Monte Carlo method introduced by ref. 71 and formalized by ref. 72 We perform the experiment during 2023 for each prior inventory and use 40 inversion ensemble members using the inversion configuration with TROPOMI $X_{CO}$ representativeness errors and 7 day optimization.

Finally, for each prior inventory, we calculate the posterior best estimates and uncertainties from the experiments described above (Extended Data Fig. 1d). The best estimate is taken to be the mean across the four different inversion configurations. The uncertainty on this estimate is taken as the square-root of the sum of the variances resulting from the different inversion configurations and Monte Carlo posterior covariance estimate. The overall best estimate is taken to be the average across the best estimates for the prior inventory ensembles and the overall uncertainty is taken to be the range of $1\sigma$ uncertainties across the three prior inventory ensembles.

We estimate posterior $CO_2$ fluxes from the posterior CO emissions using the $CO_2/CO$ emission ratios provided by the prior GFED4.1s, GFAS and QFED inventories. Each inventory has different $CO_2/CO$ emission; thus, we use the emission ratio to estimate the posterior $CO_2$ from the same inventory that was used as the prior inventory. This incorporates some uncertainty $CO_2/CO$ emission ratio into the $CO_2$ emission estimates.

### Regional masks

**Forest area.** Forest area is defined using v.6.1 of the MODIS MCD12C1 product[73]. On the basis of the type 1 majority land cover, we define forests to include the categories evergreen needleleaf forests, evergreen broadleaf forests, deciduous needleleaf forests, deciduous broadleaf forests, mixed forests, woody savannas and savannas.

**Managed land.** The map of managed lands[74] was accessed through personal communication with M. Hafer and A. Dyk (the map was only created for cartographic communication purposes). The extent of land considered managed forest in Canada for the purposes of GHG reporting to the United Nations Framework Convention on Climate Change cannot be mapped in detail. That information comes from provincial/territorial forest inventories that are not spatially explicit and cannot be mapped. Supplementary Fig. 13 shows the managed land map and the fractional managed/unmanaged for 2° × 2.5° grid cells.

## Data availability

The dataset produced for this study can be accessed at JPL Open Repository, https://doi.org/10.48577/jpl.V5GR9F.

## Code availability

The Python and Bash codes used in this study are available at Zenodo (https://doi.org/10.5281/zenodo.12709398)[75].

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

**Acknowledgements** The research carried out at the Jet Propulsion Laboratory, California Institute of Technology, was under a contract with the National Aeronautics and Space Administration. Resources supporting this work were provided by the NASA High-End Computing programme through the NASA Advanced Supercomputing Division at Ames Research Center. Authors B.B., A.C., J.L. and K.B. acknowledge the support from NASA Orbiting Carbon Observatory Science Team Program and the Carbon Monitoring System Program (grant no. NNH20ZDA001N-CMS). We acknowledge the World Climate Research Programme, which, through its Working Group on Coupled Modelling, coordinated and promoted CMIP6. We thank the climate modelling groups for producing and making available their model output, the Earth System Grid Federation (ESGF) for archiving the data and providing access and the many funding agencies who support CMIP6 and ESGF. GFAS is generated using Copernicus Atmosphere Monitoring Service Information 2020; neither the European Commission nor ECMWF is responsible for any use that may be made of the information it contains. The East Trout Lake TCCON station is funded through an infrastructure grant from the Canada Foundation for Innovation (grant no. 35278) and the Ontario Research Fund (grant no. 35278). The Park Falls TCCON site was supported by NASA (grant no. 80NSSC22K1066). We thank J. L. Laughner for guidance with the TCCON data. We thank M. Hafer and A. Dyk for providing information on Canada's managed land. And we thank L. Baskaran for help in rasterizing these data.

**Author contributions** B.B., J.L., K.W.B., M. P.-C., A.C. and S.P. conceptualized and designed the study. K.M. provided atmospheric CO production and OH estimates. G.R.v.d.W extended the GFED4.1s dataset for this experiment. D.W., P.O.W. and C.M.R. provided TCCON data. S.S. provided MERRA-2 reanalysis for the model. B.B. conducted the analysis and wrote the paper, with input from all authors.

**Competing interests** The authors declare no competing interests.

**Additional information**
**Correspondence and requests for materials** should be addressed to Brendan Byrne.

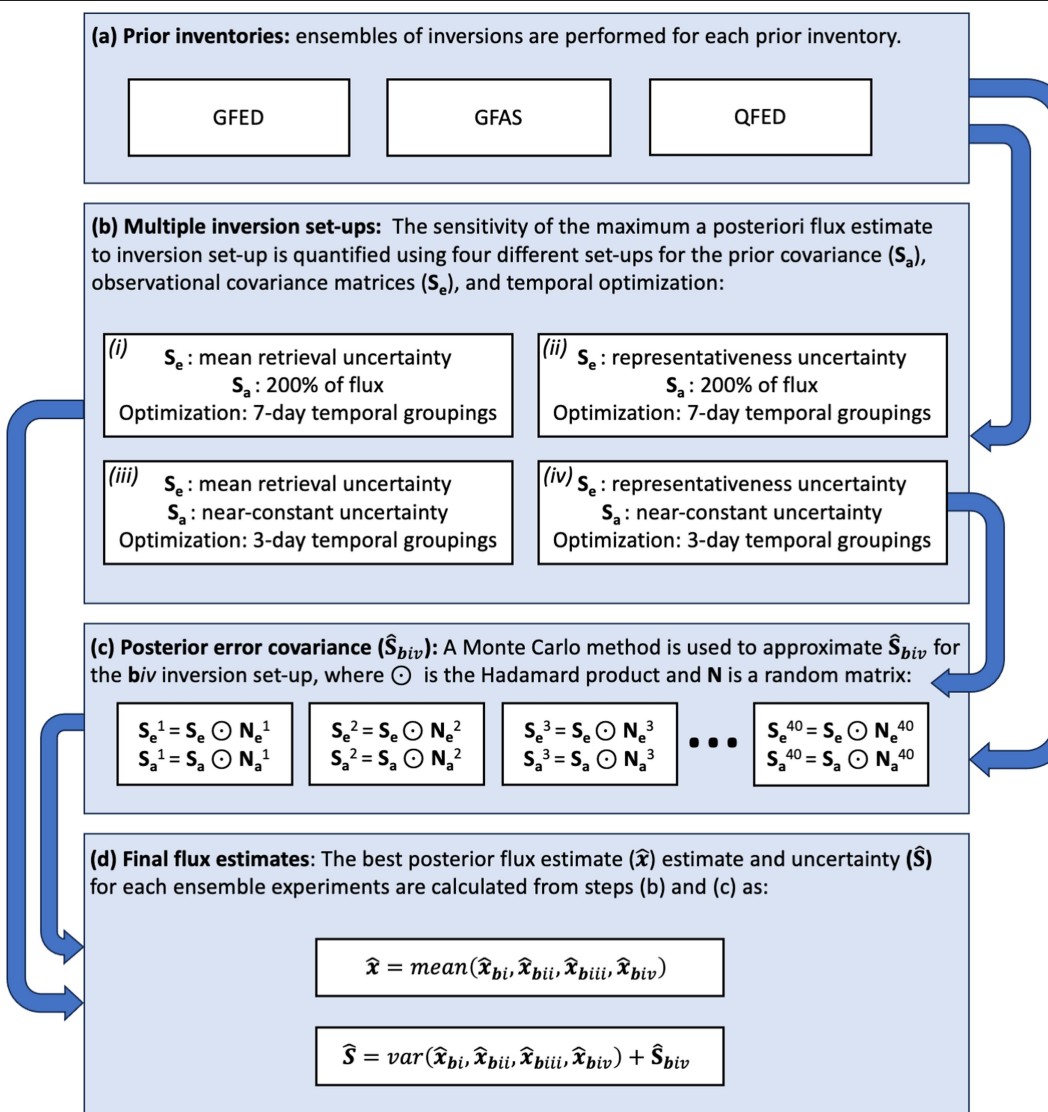

**(a) Prior inventories:** ensembles of inversions are performed for each prior inventory.

GFED    GFAS    QFED

**(b) Multiple inversion set-ups:** The sensitivity of the maximum a posteriori flux estimate to inversion set-up is quantified using four different set-ups for the prior covariance ($S_a$), observational covariance matrices ($S_e$), and temporal optimization:

*(i)* $S_e$ : mean retrieval uncertainty
$S_a$ : 200% of flux
Optimization: 7-day temporal groupings

*(ii)* $S_e$ : representativeness uncertainty
$S_a$ : 200% of flux
Optimization: 7-day temporal groupings

*(iii)* $S_e$ : mean retrieval uncertainty
$S_a$ : near-constant uncertainty
Optimization: 3-day temporal groupings

*(iv)* $S_e$ : representativeness uncertainty
$S_a$ : near-constant uncertainty
Optimization: 3-day temporal groupings

**(c) Posterior error covariance ($\hat{S}_{biv}$):** A Monte Carlo method is used to approximate $\hat{S}_{biv}$ for the **b***iv* inversion set-up, where $\odot$ is the Hadamard product and **N** is a random matrix:

$S_e^1 = S_e \odot N_e^1$
$S_a^1 = S_a \odot N_a^1$

$S_e^2 = S_e \odot N_e^2$
$S_a^2 = S_a \odot N_a^2$

$S_e^3 = S_e \odot N_e^3$
$S_a^3 = S_a \odot N_a^3$

$\cdots$

$S_e^{40} = S_e \odot N_e^{40}$
$S_a^{40} = S_a \odot N_a^{40}$

**(d) Final flux estimates:** The best posterior flux estimate ($\hat{x}$) estimate and uncertainty ($\hat{S}$) for each ensemble experiments are calculated from steps (b) and (c) as:

$$\hat{x} = mean(\hat{x}_{bi}, \hat{x}_{bii}, \hat{x}_{biii}, \hat{x}_{biv})$$

$$\hat{S} = var(\hat{x}_{bi}, \hat{x}_{bii}, \hat{x}_{biii}, \hat{x}_{biv}) + \hat{S}_{biv}$$

**Extended Data Fig. 1 | Schematic diagram of the TROPOMI XCO inversion procedure.** (a) Ensembles of inversions are performed based on three different flux inventories. (b) To quantify the sensitivity to systematic error sources, four inversions are performed that differ in observational error constraints, prior error constraints, and temporal optimization frequency. (c) Bayesian posterior error estimates are estimate for 2023 by following the Monte Carlo approach for 4D-Var of Chevalier et al.[71]. (d) The posterior best estimates are taken as the average maximum a posteriori estimate across inversion configurations while the uncertainty is taken to be the sum-of-squares of the error components estimated in (b) and (c).