## [Peer Review File · Nature]

Manuscript Title: Vast 2023 Canadian forest fire carbon emissions

Reviewer Comments & Author Rebuttals

Reviewer Reports on the Initial Version:

Referees' comments:

Referee #1 (Remarks to the Author):

Review of Brendan Byrne "Unprecedented Canadian forest fire carbon emissions during 2023" submitted to Nature

The authors calculate the amount of carbon emitted by the Canadian forest fires during 2023 using satellite-based observations of the carbon monoxide in the smoke plumes as primary source of information. This is additional, and likely more accurate, information to the currently available bottom-up emission inventories. They put the figure into context, showing that it is unprecedented in Canada and that it is of similar magnitude as the annual fossil fuel emissions of India. They go on to show that the extreme Canadian fire activity of 2023 is associated with extreme anomalies in temperature and precipitation (hot and dry), and find that similar conditions are typically found in CMIP6 climate projections after 2050. Assuming that Canada would experience similarly extreme fires much more frequently after 2050, the authors conclude, that the Canadian forests will not be able to maintain their current role as globally relevant carbon sink.

The work is overall original, significant, and suitable for publication in Nature. I expect that the conclusions will be of immediate interest to many people in several disciplines. Given the coverage of these fires in the global news and the exposure of a large fraction of the North American public to the smoke from these fire during 2023, this work may also be of interest beyond the scientific community.

The approach is generally valid and the conclusions are supported by the manuscript. However, I recommend to update the calculation and presentation regarding five key aspects:

1. According to Section 7.4, the inversions using GFAS and QFED as a priori are performed using 3-day scale factors, while the one for GFED uses 7-day scale factors. The number of degrees of freedom during any given time period is thus more than twice as large than for the GFED-based inversions. I do not expect the headline results to change significantly because of this, but it makes detailed results like the comparisons in Figs. S8 and S9 less meaningful. For example, these figures show a decrease of the standard deviation after the optimisation with GFED but an increase after the optimisations with GFAS and QFED; is this due to a different configuration of the inversion/optimisation or due to differences in the datasets? Likewise, different assumptions are made on the a priori uncertainties of small fluxes. The work would benefit from making the three inversions consistent.

2. According to Sect. 7.2.2, fire emissions are released at the model surface. This is, if I am not mistaken, not common practice because it can lead to superfluous deposition in some models. There is thus a potential that the inversions systematically overestimate the emissions. I recommend to verify that this problem does not occur in the the presented work and document this in the supplemental material.

3. According to page 12 and Fig. S6, the simulated a priori fields of CO have a smooth large-scale bias, which gets reduced by the inversion. It thus gets attributed to errors in the a priori fire emissions. However, the uniformity of the bias suggests to me that it is not due to fire emissions but some (other) model error. This is another potential source for a systematic bias in the fire emssion estimate. Such a bias needs to be quantified and corrected. (In my experience, inversions must fit the elevation of the CO concentrations in the smoke plume above the CO background instead of simply fitting CO columns. Also, plotting model-observation instead of observation-model in Fig. S6 would show the model error and be easier to discuss.)

4. It is understood that it is impossible to describe all subtleties of the algorithms of an atmospheric inversion systems in a scientific article. But Section 7.4 is too superficial. I recommend to cite a description of CMS-Flux, even if it is not peer-reviewed, and answer question like the following, which are specific to the setup used here, in the manuscript:

- How long has the 4D-Var assimilation window been selected to be?
- What motivates the selection of days for which the scale factors that are assumed to be constant?
- What justifies setting the a priority uncertainty to 200%?
- How much are the uncertainties of small fires inflated and how is this motivated?

5. The conclusions on the Canadian carbon budget implications summarised in the last two sentences of the abstract are of qualitative nature while, the other conclusions are based on a quantitative analysis. While I agree with these conclusions, I still feel that the context of the quantitative analysis gives the last two sentences of the abstract undue weight. Furthermore, I have the impression that these qualitative conclusions can be drawn from the already published work cited in Section 5. These conclusions thus appear not to be original work that is based on the presented quantitative analysis. Therefore, I recommend a major update of Section 5 that is more clear and detailed about how the authors arrive at these conclusions, and what additional insights are actually gained in this context from the presented quantitative analysis. The abstract should also be adapted correpondingly, possibly removing the last two sentences.

Statistics are used appropriately and uncertainties are treated appropriately, apart from what I wrote above.

The conclusions are robust, valid and reliable, apart from what I wrote above.

No additional experiments or data are needed, apart from what I wrote above.

The manuscript references previous literature appropriately, apart from

- missing reference to CMS-Flux, see above
- [12] Di Giuseppe et al. 2018 should be removed: It does not describe the actual implementation of

GFAS.

The manuscript is generally clear and appropriate. It could still benefit from an editorial iteration by the coauthors, though, and I list a few minor suggestions here, too:

* Abstract:

"Fire weather": I am not aware that this term is generally be understood to refer to weather with large fire danger. You may want to be more explicit.

* 1 Introduction:

"While increasing fire activity has been widespread": I am not sure that I understand what is meant. I suggest to rephrase and add a reference.

Fig. 1: "XCO": The term/quantity needs to be defined/explained.

page 4, last two sentences: Consider moving these statements to conclusions. Also, I suggest to be consistent about whether the pyrogenic emissions of carbon or inhibiting carbon uptake by changes in ecosystems due to fire is the more relevant mechanism for the carbon cycle.

* Equations 1-3:

Mathematical variables are usually set in italics. Also, why has "Z-score" a subscript in (2) but not in (3)? Also, generally, I prefer to have variables instead of words in mathematical equations, but this may be a question of journal style and I leave it to the editor.

Sect. 7.3.2:

Please motivate why you chose to treat MOPITT differently from TROPOMI by generating super-obs for one but not the other.

Referee #2 (Remarks to the Author):

This manuscript quantifies the carbon emissions from the 2023 Canadian wildfires based upon inverse modeling of satellite carbon monoxide observations.

Some discussion on combustion, flaming and smouldering and how well the approaches capture these emissions accurately would be useful context. Flaming combustion is relatively short-lived whereas smouldering can continue for weeks, months or longer especially in deep organic layers. The amount of emissions from these deep organic layers can be significant (Turetsky et al. 2015). Peat plays can play a major role in the carbon balance of forested land as peatlands can be considered legacy carbon building up over thousands of years that can be injected into the atmosphere will one deep burning fire.

There is still considerable scatter in the performance of atmospheric CO inversions (figures s8 and s9) particularly with elevated XCO mole fractions above 200 ppb. The authors comment that, in part, this is due to issues in transport models modelling the plumes. Do they see these modelling problems being resolved in the near future and what other factors may be contributing to the scatter?

The area burned numbers are preliminary estimates. The most reliable area burned data is from the National Burned Area Composite (NBAC <https://cwfis.cfs.nrcan.gc.ca/datamart/metadata/nbac>) and I am hearing preliminary numbers of 15-16 million ha of Canadian forests burned. Canadian forests can not sustain the level of burning seen in 2023. If wildfires continue to be more frequent, we will see the forest convert to shrubland or grassland. The point that Canadians forests are not a reliable sink and are more likely a carbon source due to disturbances like wildfires and insects (Mountain Pine Beetle)

For the fire and climate anomalies did you consider using components of the Canadian Forest Fire Danger Rating System in this study (Amiro et al. 2001).

Minor comments

Did you consider including October 2023 in this study. There were still some active burning in October.

Some discussion of fire management would be useful. Many wildfires in northern areas are monitored and not suppressed so they have an opportunity to grow large.

References

Amiro, B.D., Todd, J.B., Wotton, B.M., Logan, K.A., Flannigan, M.D., Stocks, B.J., Mason, J.A., Skinner, W.R., Martell, D.L. and Hirsch, K.G. 2001. Direct carbon emissions from Canadian forest fires, 1959 to 1999. *Canadian Journal of Forest Research*. 31:512-525. doi.org/10.1139/x00-197

Turetsky, M., Benscoter, B., Page, S. et al. Global vulnerability of peatlands to fire and carbon loss. *Nature Geosci* 8, 11–14 (2015). <https://doi.org/10.1038/ngeo2325>

Referee #3 (Remarks to the Author):

Review on:

Unprecedented Canadian Forest fire carbon emissions during 2023” by Brendan Byrne.

Byrne et al., 2023 employs a global inverse model system that assimilates CO satellite retrievals from MOPITT and TROMOPI to derive forest fire fluxes (CO₂+ CO) over the Canada region. Some of the main results of this study indicate that the 2023 fire events in Canada were anomalous, suggesting that the primary climate drivers of these unprecedented fires were higher than average temperatures and lower than average rainfall for Canada. To understand how these fires will impact the future, Byrne et al., 2023 analyzed 27 models from CMIP6 and found that the fire events that occurred in 2023 would likely happen again by 2050. While the study's design is good and relevant for understanding Canada's fire emissions and their future impact, there are certain ambiguities in its methodology that are not properly explained, raising concerns regarding the validity and reliability of the conclusions drawn. Below, I provide several comments that need to be addressed by the authors before this study can be published.

Section 2. Fires

Figure 1c illustrates that for the year 2023, TROPOMI-derived fire carbon emissions reduced the range of discrepancy among different global inventory fire products (GFED, QFAS, QFED) by 73%. While these results are good because the means of these are closer, it is essential to understand the posterior uncertainties associated with each of these three individual TROPOMI fire estimates. Since inversion using QFAS and QFED was only conducted for the year 2023, a Monte Carlo ensemble approach could be employed to consider various realizations of prior uncertainties. It remains unclear why a uniform prior uncertainty of 200% was assumed for GFED and how much the uncertainties in QFAS and QFED were inflated in the trial-and-error experiment mentioned in the method section.

Section 3: 2023 Climate anomalies

In section 3, it is unclear why the authors only show the spatial anomaly map of fire CO₂+CO emissions from the GFED4.1s database and do not show the results of the TROPOMI, MOPITT (CO₂ + CO) inversion.

While the authors focus on explaining Canada's forest climate anomalies, they do not attempt to explain how well TROPOMI flux anomalies are correlated to these climate variables. Are prior fluxes (e.g., GFED shown in Fig 2 panels di and dii) used in the inversion better correlated to the climate variables than the TROPOMI-derived fluxes? It is possible to create a map of correlations that shows how well the prior and posterior are correlated to the climate variables.

In Figure 2, why does the author not show the spatial anomalies and accumulative fire emissions from TROPOMI-derived fluxes? Isn't this estimate one important result of the study as stated in the abstract?

Section 4: Fires and Climate

In section 4, the author explores the relationship between climate and fire emissions anomalies for Canadian forests. For this analysis, they computed a z-score for different climate data (e.g. temperature and precipitation) relative to the 2003-2022 period. While this analysis seems appropriate to examine how the current climate has changed relative to past conditions, the selection of the CMIP6 models to understand future climate conditions is not entirely clear. The author selected 27 models from CMIP6 to calculate the z-score, but what were the criteria for this selection? Are all these models together reasonably reproducing the observed annual cycles of temperature and rainfall across forest areas in Canada? Recent studies (e.g., Palmer et al, 2022) suggest that CMIP6 model performance can vary significantly across different regions across the world, therefore a selection of the model should be accounted for in regional-scale weather fire studies. What if not all the models performed well for Canada, what if the multi-model mean of these 27 is significantly impacted by outlier models, presenting unrealistic estimates of rainfall and temperature. Fig 3 suggests that by 2050 similar weather conditions could lead to similar forest fires that occurred in 2023, but this is not entirely correct, and this event could happen in a shorter period. I highlight this point because one of the main conclusions of this study is that “CMIP6 climate models project that the temperatures of 2023 will become the norm by the 2050s”.

Section 5: Canadian Carbon Budget Implications

In section 5, the authors highlight the importance of considering fire emissions from unmanaged lands in Canada's carbon budget (e.g., Canada National GHG inventory report), but they fail to provide information on these unmanaged fires. Given that this section discusses the implications for Canada's national carbon budget, I would have expected the authors to differentiate between forest fire emissions for managed and unmanaged forest land. For this purpose, utilizing the forest mask from the National Forest Carbon Monitoring, Accounting, and Reporting System (NFCMARS) would be ideal for this straightforward quantification(see: <https://natural-resources.canada.ca/climate-change-adapting-impacts-and-reducing-emissions/climate-change-impacts-forests/carbon-accounting/inventory-and-land-use-change/13111>). Could the authors explain why they chose to use a global dataset from MODIS instead of a more regional one? Isn't this a regional study? MODIS does not appear to be the most suitable product for delineating information about managed and unmanaged forest land. How different are these two products in terms of forest area? Having this information would likely capture the attention of the Canadian government and policymakers regarding how they track their national emissions, and call for a change to the implementation of the Paris Agreement's reporting mechanisms.

Methods

7.3 CO retrievals

7.3.1 TROPOMI

How were the uncertainties of the dry-air mole fractions of CO from TROPOMI calculated? The description of how TROPOMI observations were averaged in this section is vague and requires more detail. What is the temporal resolution of the super-observations? It would be good to include some equations here.

7.3.2 MOPITT

It is not clear how the assimilation of MOPITT satellite retrieval was performed. What does the author mean by stating that the inverse system assimilates 'individual observations'? How does this product differ from TROPOMI, which cannot be averaged in space and time?

7.4. Atmospheric CO inversion

The description of the methodology is quite ambiguous; the authors do not clearly explain some important components of the CMS-Flux-TROPOMI inversion system. What are the references to this system?

What is the chemical transport model used in the inversion? Is it the Geos Chem model? Is the chemical production of CO coming from the oxidation of methane and non-methane volatile organic compounds considered?

For any inverse flux system, errors in the model transport need to be estimated; otherwise, the resulting errors between the data-model mismatch might prevent observations from being used effectively in the 4D-Var. How were the error 'uncertainties' in the model calculated and added to TROPOMI observational uncertainties? This also applies to the description of the prior uncertainties. For example, in Section 7.4, the authors state that "Prior uncertainty emissions are assumed to be proportional to the emissions, with a scale factor uncertainty of 200%". Could the authors explain the basis of this assumption? Is there any reference for this, or analysis made that was forgotten to include in the text? I don't understand what is meant to say that uncertainties are inflated when CO₂ emissions are smaller. How was the trial-and-error test performed?

Another point that is not clear is how the contribution of the prior was calculated to infer the posterior CO fluxes. In general, the inversion approach described is difficult to understand and not well explained. Please also provide information about how the CO₂/CO emission ratios were calculated. A recent paper published by Peiro et al. in 2023 clearly provides information on how this was estimated. What variables are optimized in the control vector?

Supplementary information

Is there any reason why the author did not provide information about the validation of the posterior CO fields against the assimilated MOPITT XCO retrievals, and TCCON?

In academic writing, it would be better to avoid using the phrase "too low" and instead opt for a more precise and formal expression. You could rephrase the sentence to something like:

"The prior inventories indicate significant positive biases, suggesting that the model-simulated XCO levels are underestimated."

Specific comments on the text:

In Section 2, paragraph 1, Line 5, the authors says that bottom-up inventories vary significantly in global and regional trace gas and aerosol emission estimates. I would add that the top-down approaches also vary significantly.

Throughout the text, I found that Supplementary Figures are not well referenced. For example, in this section of the text: 'Western Quebec (49°–55° N, 72°–80° W), which is typically relatively wet (Fig. S3),' the authors reference Fig. S3, but this Figure has 3 panels. It would be clearer for the reader if they specified which panel they're referring to. I'm assuming they mean Fig. S3a?

References:

Palmer, T. E., McSweeney, C. F., Booth, B. B. B., Priestley, M. D. K., Davini, P., Brunner, L., Borchert, L., and Menary, M. B.: Performance-based sub-selection of CMIP6 models for impact assessments in Europe, *Earth Syst. Dynam.*, 14, 457–483, <https://doi.org/10.5194/esd-14-457-2023>, 2023.

Peiro, H., Crowell, S., and Moore III, B.: Optimizing 4 years of CO₂ biospheric fluxes from OCO-2 and in situ data in TM5: fire emissions from GFED and inferred from MOPITT CO data, *Atmos. Chem. Phys.*, 22, 15817–15849, <https://doi.org/10.5194/acp-22-15817-2022>, 2022.

Author Rebuttals to Initial Comments:

We thank the referees for their thoughtful comments. We have addressed the issues brought up by the referees in our revised manuscript, with point-by-point responses below. Reviewer comments are in black and our responses in blue.

One common theme across the reviews was a desire for more a complete discussion of the sensitivity of our inversion results to error sources. In response, we have re-designed our experimental procedure to better track uncertainties in the inversion results. In revised Sec. 7.4, you will see that we now perform an ensemble of inversions to track both systematic sources of error due to inversion configuration and Bayesian sources of errors in the posterior fluxes. We have revised to main text to reflect this new set of experiments, but the qualitative results of our analysis are not changed.

Another common theme was to refine the discussion about the Canadian carbon budget. We have also revised Sec. 5 to compare fire emissions over managed forests to the Canadian National Greenhouse Gas Inventory.

Referees' comments:

Referee #1 (Remarks to the Author):

Review of Brendan Byrne "Unprecedented Canadian forest fire carbon emissions during 2023" submitted to Nature

The authors calculate the amount of carbon emitted by the Canadian forest fires during 2023 using satellite-based observations of the carbon monoxide in the smoke plumes as primary source of information. This is additional, and likely more accurate, information to the currently available bottom-up emission inventories. They put the figure into context, showing that it is unprecedented in Canada and that it is of similar magnitude as the annual fossil fuel emissions of India. They go on to show that the extreme Canadian fire activity of 2023 is associated with extreme anomalies in temperature and precipitation (hot and dry), and find that similar conditions are typically found in CMIP6 climate projections after 2050. Assuming that Canada would experience similarly extreme fires much more frequently after 2050, the authors conclude, that the Canadian forests will not be able to maintain their current role as globally relevant carbon sink.

The work is overall original, significant, and suitable for publication in Nature. I expect that the conclusions will be of immediate interest to many people in several disciplines. Given the coverage of these fires in the global news and the exposure of a large fraction of the North American public to the smoke from these fire during 2023, this work may also be of interest beyond the scientific community.

The approach is generally valid and the conclusions are supported by the manuscript. However, I recommend to update the calculation and presentation regarding five key aspects:

1. According to Section 7.4, the inversions using GFAS and QFED as a priori are performed using 3-day scale factors, while the one for GFED uses 7-day scale factors. The number of degrees of freedom during any given time period is thus more than twice as large than for the GFED-based inversions. I do not

expect the headline results to change significantly because of this, but it makes detailed results like the comparisons in Figs. S8 and S9 less meaningful. For example, these figures show a decrease of the standard deviation after the optimisation with GFED but an increase after the optimisations with GFAS and QFED; is this due to a different configuration of the inversion/optimization or due to differences in the datasets? Likewise, different assumptions are made on the a priori uncertainties of small fluxes. The work would benefit from making the three inversions consistent.

In the revised manuscript we now employ an ensemble of four inversion configurations for each inventory. Figures S8-9 (now Figs S9-10) have been revised to show the mean across this ensemble, while tables S1-3 have been added to give the statistics of the observation minus model mismatch for each inversion.

2. According to Sect. 7.2.2, fire emissions are released at the model surface. This is, if I am not mistaken, not common practice because it can lead to superfluous deposition in some models. There is thus a potential that the inversions systematically overestimate the emissions. I recommend to verify that this problem does not occur in the presented work and document this in the supplemental material.

We agree that fire emissions are often released at an estimated plume height as an attempt to account for pyro-convective motions that are not modeled. In the revised manuscript, we have performed a series of forward simulations with fire CO released at the plume injection height as estimated by IS4FIRES (<https://en.ilmatieteenlaitos.fi/information-system-on-vegetation-fires>). We find that the difference in mean bias against observations to be 0-5 ppb, which is generally considerably smaller than the prior/posterior differences, thus we believe our flux estimates are robust to injection height.

3. According to page 12 and Fig. S6, the simulated a priori fields of CO have a smooth large-scale bias, which gets reduced by the inversion. It thus gets attributed to errors in the a priori fire emissions. However, the uniformity of the bias suggests to me that it is not due to fire emissions but some (other) model error. This is another potential source for a systematic bias in the fire emission estimate. Such a bias needs to be quantified and corrected. (In my experience, inversions must fit the elevation of the CO concentrations in the smoke plume above the CO background instead of simply fitting CO columns. Also, plotting model-observation instead of observation-model in Fig. S6 would show the model error and be easier to discuss.)

To clarify, we optimize the net surface-atmosphere CO flux, rather than just fire CO. Posteriori fluxes are then decomposed into fossil fuel, fire, and Biogenic sources using the fractional contribution to each gridcell. This is a commonly used approach within the CO inversion community. This is described in Sec 7.4.

Regarding the large-scale bias, the positive observation minus model pattern derives from the fact that prior fluxes are insufficient to sustain observed CO mole fractions. This is a well-documented problem in the CO budget (see Table S4 of Zeng et al., 2019). This budget may partially result from an underestimate of small fires in fire inventories, particularly in the tropics, which more recent estimates suggesting increased global fire CO emissions (Chen et al., 2023; Ramo et al., 2021).

We do examine the fit of fire-impacted CO observations against the two TCCON sites (Figures S9-10). We find that the posterior fluxes show a slope that is much closer to 1:1 than the prior fluxes, especially for <200 ppb. This suggests that the inversion is both correcting for the large-scale bias and also for the regional fire CO emissions.

References

- Zheng et al.: Global atmospheric carbon monoxide budget 2000–2017 inferred from multi-species atmospheric inversions, *Earth Syst. Sci. Data*, 11, 1411–1436, <https://doi.org/10.5194/essd-11-1411-2019>, 2019.
- Ramo et al.: African burned area and fire carbon emissions are strongly impacted by small fires undetected by coarse resolution satellite data, *PNAS*, 118 (9) e2011160118 <https://doi.org/10.1073/pnas.2011160118>, 2021.
- Chen, Y., Hall, J., van Wees, D., Andela, N., Hantson, S., Giglio, L., van der Werf, G. R., Morton, D. C., and Randerson, J. T.: Multi-decadal trends and variability in burned area from the fifth version of the Global Fire Emissions Database (GFED5), *Earth Syst. Sci. Data*, 15, 5227–5259, <https://doi.org/10.5194/essd-15-5227-2023>, 2023.

4. It is understood that it is impossible to describe all subtleties of the algorithms of an atmospheric inversion systems in a scientific article. But Section 7.4 is too superficial. I recommend to cite a description of CMS-Flux, even if it is not peer-reviewed, and answer question like the following, which are specific to the setup used here, in the manuscript:

- How long has the 4D-Var assimilation window been selected to be?
- What motivates the selection of days for which the scale factors that are assumed to be constant?
- What justifies setting the a priori uncertainty to 200%?
- How much are the uncertainties of small fires inflated and how is this motivated?

Please see revised Sec. 7.4. We have added these details when describing the revised ensembles of inversions.

5. The conclusions on the Canadian carbon budget implications summarized in the last two sentences of the abstract are of qualitative nature while, the other conclusions are based on a quantitative analysis. While I agree with these conclusions, I still feel that the context of the quantitative analysis gives the last two sentences of the abstract undue weight. Furthermore, I have the impression that these qualitative conclusions can be drawn from the already published work cited in Section 5. These conclusions thus appear not to be original work that is based on the presented quantitative analysis. Therefore, I recommend a major update of Section 5 that is more clear and detailed about how the authors arrive at these conclusions, and what additional insights are actually gained in this context from the presented quantitative analysis. The abstract should also be adapted correspondingly, possibly removing the last two sentences.

The final two sentences of the abstract state: *“Although extreme relative to the historical record, climate projections indicate that these temperatures will be typical during the 2050s, even under a moderate climate mitigation scenario (SSP2-4.5). Such conditions are very likely to drive increased fire activity and suppress carbon uptake by Canadian forests, calling into question the long-term durability of these forests as a carbon sink.”*

We believe the first sentence and first half of the second sentence are well supported by the analyses in Section 4. This section demonstrates warming over Canadian forests from analyzing CMIP6 output. It also demonstrates the relationship between fire, temperature and precipitation over Canadian boreal forests from analyzing GFED4.1s and MERRA2 data, showing that above average temperatures are correlated with fire emissions. We then connect this relationship to observed and projected increases in fire activity from previous studies, which also show a suppressive impact on carbon accumulation due to

fire. We agree that the second half of the second sentence is more speculative, but believe that this is an important question to highlight given both the unprecedented nature of carbon loss during the 2023 fires and the projected warming in the region.

We have also revised Section 5 with a more detailed comparison of our results to the Canadian National Greenhouse Gas Inventory.

Statistics are used appropriately and uncertainties are treated appropriately, apart from what I wrote above.

The conclusions are robust, valid and reliable, apart from what I wrote above.

No additional experiments or data are needed, apart from what I wrote above.

The manuscript references previous literature appropriately, apart from

- missing reference to CMS-Flux, see above

Added

- [12] Di Giuseppe et al. 2018 should be removed: It does not describe the actual implementation of GFAS.

This reference has been removed

The manuscript is generally clear and appropriate. It could still benefit from an editorial iteration by the coauthors, though, and I list a few minor suggestions here, too:

* Abstract:

"Fire weather": I am not aware that this term is generally be understood to refer to weather with large fire danger. You may want to be more explicit.

"fire weather" has been replaced with "hot-dry weather"

* 1 Introduction:

"While increasing fire activity has been widespread": I am not sure that I understand what is meant. I suggest to rephrase and add a reference.

Changed to "While more frequent fires have been widespread"

Fig. 1: "XCO": The term/quantity needs to be defined/explained.

Added description "dry-air mole fractions of CO"

page 4, last two sentences: Consider moving these statements to conclusions. Also, I suggest to be

consistent about whether the pyrogenic emissions of carbon or inhibiting carbon uptake by changes in ecosystems due to fire is the more relevant mechanism for the carbon cycle.

These sentences were redundant with existing text in the conclusions, so we have removed.

* Equations 1-3:

Mathematical variables are usually set in italics. Also, why has "Z-score" a subscript in (2) but not in (3)? Also, generally, I prefer to have variables instead of words in mathematical equations, but this may be a question of journal style and I leave it to the editor.

We have added the subscript year in both equations for consistency

Sect. 7.3.2:

Please motivate why you chose to treat MOPITT differently from TROPOMI by generating super-obs for one but not the other.

The footprint of MOPITT CO retrievals ($22 \times 22 \text{ km}^2$) is much coarser than TROPOMI retrievals ($3 \times 7 \text{ km}^2$). This has been clarified in the revised text.

Referee #2 (Remarks to the Author):

This manuscript quantifies the carbon emissions from the 2023 Canadian wildfires based upon inverse modeling of satellite carbon monoxide observations.

Some discussion on combustion, flaming and smouldering and how well the approaches capture these emissions accurately would be useful context. Flaming combustion is relatively short-lived whereas smouldering can continue for weeks, months or longer especially in deep organic layers. The amount of emissions from these deep organic layers can be significant (Turetsky et al. 2015). Peat plays can play a major role in the carbon balance of forested land as peatlands can be considered legacy carbon building up over thousands of years that can be injected into the atmosphere will one deep burning fire.

We have added the sentence "A strength of this approach is that integrates emissions from combustion, flaming and smoldering potentially capturing net emissions." To emphasize that the resulting CO emissions from all these phases will be captured. However, we do not focus on characterizing the importance of these different fire types to total emissions, as this is challenging through top-down methods. Instead, emission factors are derived in the bottom-up models we use. They are based on a mixture of ground and mostly airborne measurement over a wide range of fire types and stages.

There is still considerable scatter in the performance of atmospheric CO inversions (figures s8 and s9) particularly with elevated XCO mole fractions above 200 ppb. The authors comment that, in part, this is due to issues in transport models modelling the plumes. Do they see these modelling problems being resolved in the near future and what other factors may be contributing to the scatter?

The representation of atmospheric transport is a long-standing problem, and is likely to remain a major source of error for the foreseeable future. In the revised text, we have performed sensitivity tests where

the CO emissions are released at the injection height of a plume rise model (IS4FIRES). We find that the mismatch against TCCON sites remains similar. We believe that transport error is the dominant source of mismatch.

The area burned numbers are preliminary estimates. The most reliable area burned data is from the National Burned Area Composite (NBAC <https://cwfis.cfs.nrcan.gc.ca/datamart/metadata/nbac>) and I am hearing preliminary numbers of 15-16 million ha of Canadian forests burned. Canadian forests can not sustain the level of burning seen in 2023. If wildfires continue to be more frequent, we will see the forest convert to shrubland or grassland. The point that Canadians forests are not a reliable sink and are more likely a carbon source due to disturbances like wildfires and insects (Mountain Pine Beetle)

It appears that the 2023 fire season dataset is still not available.

For the fire and climate anomalies did you consider using components of the Canadian Forest Fire Danger Rating System in this study (Amiro et al. 2001).

Thank you for sharing this resource, we were not aware of this when performing our analysis. We believe this is a valuable resource, but believe the paper has sufficient climate information in its current form, but will keep this in mind for future work.

Minor comments

Did you consider including October 2023 in this study. There were still some active burning in October.

Although fire activity continued through October, it was at a much lower intensity. The data below from CAMS GFAS shows the estimated fire emissions during Oct – early Dec.

GFASv1.2 daily total cumulative carbon emissions since 1 January (right) for Canada. Source: CAMS

Some discussion of fire management would be useful. Many wildfires in northern areas are monitored and not suppressed so they have an opportunity to grow large.

Please find a discussion of this added to the newly expanded Sec. 5 on the Canadian carbon budget:

The role of Canada's fire management strategy in managing fire carbon emissions also deserved some discussion. Fire management strategies require balancing a number of considerations, including socioeconomic costs, ecological impacts, and carbon emissions. Canada's current strategy adopts a risk-based approach, where decisions on whether or not to suppress fires are made on a fire-by-fire basis [38], with differing priorities across provinces and territories. Understanding how fire regimes will change with climate change is thus of high importance, for future decision criteria and costing.

[38] Tymstra, C., Stocks, B.J., Cai, X., Flannigan, M.D.: *Wildfire management in Canada: Review, challenges and opportunities. Progress in Disaster Science 5, 100045 (2020)*
<https://doi.org/10.1016/j.pdisas.2019.100045>

References

Amiro, B.D., Todd, J.B., Wotton, B.M., Logan, K.A., Flannigan, M.D., Stocks, B.J., Mason, J.A., Skinner, W.R., Martell, D.L. and Hirsch, K.G. 2001. Direct carbon emissions from Canadian forest fires, 1959 to 1999. *Canadian Journal of Forest Research*. 31:512-525. doi.org/10.1139/x00-197

Turetsky, M., Benscoter, B., Page, S. et al. Global vulnerability of peatlands to fire and carbon loss. *Nature Geosci* 8, 11–14 (2015). <https://doi.org/10.1038/ngeo2325>

Referee #3 (Remarks to the Author):

Review on:

Unprecedented Canadian Forest fire carbon emissions during 2023” by Brendan Byrne.

Byrne et al., 2023 employs a global inverse model system that assimilates CO satellite retrievals from MOPITT and TROMOPI to derive forest fire fluxes (CO₂+ CO) over the Canada region. Some of the main results of this study indicate that the 2023 fire events in Canada were anomalous, suggesting that the primary climate drivers of these unprecedented fires were higher than average temperatures and lower than average rainfall for Canada. To understand how these fires will impact the future, Byrne et al., 2023 analyzed 27 models from CMIP6 and found that the fire events that occurred in 2023 would likely happen again by 2050. While the study's design is good and relevant for understanding Canada's fire emissions and their future impact, there are certain ambiguities in its methodology that are not properly explained, raising concerns regarding the validity and reliability of the conclusions drawn. Below, I provide several comments that need to be addressed by the authors before this study can be published.

Section2. Fires

Figure 1c illustrates that for the year 2023, TROPOMI-derived fire carbon emissions reduced the range of discrepancy among different global inventory fire products (GFED, QFAS, QFED) by 73%. While these results are good because the means of these are closer, it is essential to understand the posterior uncertainties associated with each of these three individual TROPOMI fire estimates. Since inversion

using QFAS and QFED was only conducted for the year 2023, a Monte Carlo ensemble approach could be employed to consider various realizations of prior uncertainties. It remains unclear why a uniform prior uncertainty of 200% was assumed for GFED and how much the uncertainties in QFAS and QFED were inflated in the trial-and-error experiment mentioned in the method section.

In the revised manuscript, we have performed a series of additional inversion analyses to better quantify uncertainties in the posterior fluxes. For each inventory, we now include an ensemble of inversions that differ in assumed prior errors, observational errors, and temporal optimization. We have also performed a Monte Carlo analysis to approximate the Bayesian posterior uncertainty. These two uncertainty components are then combined in quadrature to yield our total uncertainty estimate.

Section 3: 2023 Climate anomalies

In section 3, it is unclear why the authors only show the spatial anomaly map of fire CO₂+CO emissions from the GFED4.1s database and do not show the results of the TROPOMI, MOPPIT (CO₂ + CO) inversion.

This is because the GFED4.1s results can be shown for the entire 20-year period at 0.5x0.625 resolution, while the TROPOMI inversions are at coarser (2x2.5) spatial resolution and only cover five years (2019-2023) and MOPPIT results only cover 2010-2021. We have clarified this in the caption and added a supplementary figure comparing the spatial patterns of the prior and posterior fluxes.

Figure 2 caption: “...Note that GFED4.1s is shown instead of the inversion results because those are at a coarser spatial resolution and cover a shorter time period, maps of prior and posterior fire emissions are shown in Fig. S10”

Figure S13. Maps of May–Sep 2023 CO+CO₂ fire emissions at 2° × 2.5° spatial resolution. Panels show prior (a) GFED4.1s, (b) GFAS, and (c) QFED and posterior emissions employing (d) GFED4.1s, (e) GFAS, and (f) QFED.

While the authors focus on explaining Canada’s forest climate anomalies, they do not attempt to explain how well TROPOMI flux anomalies are correlated to these climate variables. Are prior fluxes (e.g., GFED shown in Fig 2 panels di and dii) used in the inversion better correlated to the climate variables than the TROPOMI-derived fluxes? It is possible to create a map of correlations that shows how well the prior and posterior are correlated to the climate variables.

Figure S13 (shown above) shows the spatial distribution of fire emissions for the prior and posterior estimates. The spatial distribution of fluxes is similar for both prior and posterior estimates. We investigate the relationship of fire with climate using the GFED4.1s inventory so that it can be conducted at high spatial resolution (0.5 x 0.625 versus 2 x 2.5) and over a longer time period (2003-2023).

In Figure 2, why does the author not show the spatial anomalies and accumulative fire emissions from TROPOMI-derived fluxes? Isn't this estimate one important result of the study as stated in the abstract?

The spatial pattern is now shown as a supplementary figure (Fig. S13).

Section 4: Fires and Climate

In section 4, the author explores the relationship between climate and fire emissions anomalies for Canadian forests. For this analysis, they computed a z-score for different climate data (e.g. temperature and precipitation) relative to the 2003-2022 period. While this analysis seems appropriate to examine how the current climate has changed relative to past conditions, the selection of the CMIP6 models to understand future climate conditions is not entirely clear. The author selected 27 models from CMIP6 to calculate the z-score, but what were the criteria for this selection? Are all these models together reasonably reproducing the observed annual cycles of temperature and rainfall across forest areas in Canada? Recent studies (e.g., Palmer et al, 2022) suggest that CMIP6 model performance can vary significantly across different regions across the world, therefore a selection of the model should be accounted for in regional-scale weather fire studies. What if not all the models performed well for Canada, what if the multi-model mean of these 27 is significantly impacted by outlier models, presenting unrealistic estimates of rainfall and temperature. Fig 3 suggests that by 2050 similar weather conditions could lead to similar forest fires that occurred in 2023, but this is not entirely correct, and this event could happen in a shorter period. I highlight this point because one of the main conclusions of this study is that "CMIP6 climate models project that the temperatures of 2023 will become the norm by the 2050s".

To clarify, we examine the ensemble median (insensitive to outliers), not mean. We have added some clarifications to the text emphasizing we use the ensemble median.

We agree that there is considerable uncertainty in projected T2M and precipitation from the CMIP6 ensemble. However, investigating the performance of individual CMIP6 models over Canada is beyond the scope of this study. In response to the reviewer's concern, we have added a supplementary figure showing the distribution of CMIP6 models from 1980 through 2100 (see below). The ensemble members included can be found here: <https://climate-scenarios.canada.ca/?page=cmip6-model-list>, we have also added this link to the methods.

Figure S11. Box-and-whisker plots of the decadal mean CMIP6 cumulative precipitation (Jan-Sep) and mean T2M (May-Sep) over Canadian boreal forests. The black line shows the ensemble median, shaded grey area shows the interquartile range, and error bars show the 5-95% range. The decadal means and individual years of the CPC Global Unified Gauge-Based Analysis of Daily Precipitation data and MERRA-2 T2M data are shown in red. The MERRA-2 T2M is shown to correspond closely with the CMIP6 ensemble median while the Precipitation data is around the 25th percentile of the CMIP6 ensemble, suggesting most CMIP6 models overestimate precipitation over the Canadian boreal forests.

Section 5: Canadian Carbon Budget Implications

In section 5, the authors highlight the importance of considering fire emissions from unmanaged lands in Canada's carbon budget (e.g., Canada National GHG inventory report), but they fail to provide information on these unmanaged fires. Given that this section discusses the implications for Canada's national carbon budget, I would have expected the authors to differentiate between forest fire emissions for managed and unmanaged forest land. For this purpose, utilizing the forest mask from the National Forest Carbon Monitoring, Accounting, and Reporting System (NFCMARS) would be ideal for this straightforward quantification (see: <https://natural-resources.canada.ca/climate-change/adapting-impacts-and-reducing-emissions/climate-change-impacts-forests/carbon-accounting/inventory-and-land-use-change/13111>). Could the authors explain why they chose to use a global dataset from MODIS instead of a more regional one? Isn't this a regional study? MODIS does not appear to be the most suitable product for delineating information about managed and unmanaged forest land. How different are these two products in terms of forest area? Having this information would likely capture the attention of the Canadian government and policymakers regarding how they track their national emissions, and call for a change to the implementation of the Paris Agreement's reporting mechanisms.

Maps distinguishing between managed and unmanaged lands are generally not available for most countries. Recent studies looking at GHG emissions have emphasized how the absence of these data make top-down vs bottom-up comparisons challenging (Byrne et al., 2023; Deng et al., 2022). The mask posted on the referenced website was solely created for cartographic communication purposes and is not the data that Canada actually uses for GHG reporting purposes (personal communication, Mark Hafer). The extent of land considered managed forest in Canada for the purposes of GHG reporting to

the UNFCCC cannot be mapped in detail. That information comes from provincial/territorial forest inventories that are not spatially explicit and can't be mapped.

However, we were able to obtain a mask of managed / unmanaged lands (see below). In the revised manuscript, we have revised Sec. 5 to discuss the implications for the carbon emissions and removals from Canadian managed lands.

Fig. S12 (a) Managed and unmanaged lands within Canada. Fraction of inversion model $2^\circ \times 2.5^\circ$ grid cells that are (a) managed or (b) unmanaged lands.

Methods

7.3 CO retrievals

7.3.1 TROPOMI

How were the uncertainties of the dry-air mole fractions of CO from TROPOMI calculated? The description of how TROPOMI observations were averaged in this section is vague and requires more detail. What is the temporal resolution of the super-observations? It would be good to include some equations here.

We have provided more details on this calculation in the revised manuscript:

“Retrieved CO total column densities are then converted to dry-air mole fractions of CO (XCO) using the dry-air surface pressure and hypsometric equation. The column averaging kernel is similarly converted to mole-fraction space. Individual retrievals (quality flag ≥ 0.5) from each orbit are aggregated into super-observations using the model grid ($2^\circ \times 2.5^\circ$).”

The retrieval uncertainty on super-observations is taken to be the mean uncertainty on all retrievals within a given super-observations. This approach is used because systematic errors may exist between retrievals, such that assuming random errors would underestimate the true retrieval error. For assimilation into CMS-Flux, we calculate observational errors that incorporate error in the atmospheric transport model. For this, we follow the approach of Heald et al. [52]. First, we perform a forward model simulation with the prior fluxes for 2019–2023. Then we take the observational uncertainty to be the standard deviation between the simulated and real TROPOMI super-obs over a moving window of 30° latitude, 30° longitude, and 30 days (across all years). The uncertainties estimated using this approach range over 3.5-14.3 ppb (5–95 percentiles), while retrieval errors range over 1.4-4.9 ppb. Thus, the observational errors are dominated by representativeness errors.”

7.3.2 MOPITT

It is not clear how the assimilation of MOPITT satellite retrieval was performed. What does the author mean by stating that the inverse system assimilates 'individual observations'? How does this product differ from TROPOMI, which cannot be averaged in space and time?

The footprint of MOPITT CO retrievals ($22 \times 22 \text{ km}^2$) is much coarser than TROPOMI retrievals ($3 \times 7 \text{ km}^2$). This has been clarified in the revised text.

7.4. Atmospheric CO inversion

The description of the methodology is quite ambiguous; the authors do not clearly explain some important components of the CMS-Flux-TROPOMI inversion system. What are the references to this system?

We have added references (see below) and a more extensive description of the revised inversion methodology.

“We perform a series of CO inversion analyses using the NASA Carbon Monitoring System-Flux (CMS-Flux) atmospheric inversion system. This inversion model is descended from the GEOS-Chem adjoint model [58] and has been employed for CO₂ [59, 60] and CO inversion analyses [61].”

[58] Henze, D.K., Hakami, A., Seinfeld, J.H.: Development of the adjoint of geos-chem. Atmospheric Chemistry and Physics 7(9), 2413–2433 (2007) <https://doi.org/10.5194/acp-7-2413-2007>

[59] Liu, J., Bowman, K.W., Lee, M., Henze, D.K., Bousserez, N., Brix, H., Collatz, G.J., Menemenlis, D., Ott, L., Pawson, S., Jones, D.B.A., Nassar, R.: Carbon monitoring system flux estimation and attribution: impact of ACOS-GOSAT XCO₂ sampling on the inference of terrestrial biospheric sources and sinks. Tellus B: Chemical and Physical Meteorology 66(1), 22486 (2014) <https://doi.org/10.3402/tellusb.v66.22486>

[60] Liu, J., Baskaran, L., Bowman, K., Schimel, D., Bloom, A.A., Parazoo, N.C., Oda, T., Carroll, D., Menemenlis, D., Joiner, J., Commane, R., Daube, B., Gatti, L.V., McKain, K., Miller, J., Stephens, B.B., Sweeney, C., Wofsy, S.: Carbon monitoring system flux net biosphere exchange 2020 (cms-flux nbe 2020). Earth SystemScience Data 13(2), 299–330 (2021) <https://doi.org/10.5194/essd-13-299-2021>

[61] Byrne, B., Liu, J., Lee, M., Yin, Y., Bowman, K.W., Miyazaki, K., Norton, A.J., Joiner, J., Pollard, D.F., Griffith, D.W., et al.: The carbon cycle of southeast australia during 2019–2020: Drought, fires, and subsequent recovery. AGU Advances 2(4), 2021–000469 (2021)

What is the chemical transport model used in the inversion? Is it the Geos Chem model? Is the chemical production of CO coming from the oxidation of methane and non-methane volatile organic compounds considered?

Yes, it is derived from the GEOS-Chem adjoint model, please see revised methods. Atmospheric CO production is prescribed using the estimates from the MOMO-Chem data assimilation system described in Sec. 7.2.3.

For any inverse flux system, errors in the model transport need to be estimated; otherwise, the resulting errors between the data-model mismatch might prevent observations from being used effectively in the 4D-Var. How were the error 'uncertainties' in the model calculated and added to TROPOMI observational uncertainties? This also applies to the description of the prior uncertainties. For example, in Section 7.4, the authors state that "Prior uncertainty emissions are assumed to be proportional to the emissions, with a scale factor uncertainty of 200%". Could the authors explain the basis of this assumption? Is there any reference for this, or analysis made that was forgotten to include in the text? I don't understand what is meant to say that uncertainties are inflated when CO₂ emissions are smaller. How was the trial-and-error test performed?

Please see the revised methods where we impose several different prior and observation error parameterizations. The uncertainties on prior fire emissions are themselves very uncertain, thus we impose several assumptions, including 200% uncertainty and uniform uncertainty. The differences in maximum a posterior estimate resulting from these assumptions is then incorporated into our uncertainty on top-down emission estimates.

Another point that is not clear is how the contribution of the prior was calculated to infer the posterior CO fluxes. In general, the inversion approach described is difficult to understand and not well explained. Please also provide information about how the CO₂/CO emission ratios were calculated. A recent paper published by Peiro et al. in 2023 clearly provides information on how this was estimated. What variables are optimized in the control vector?

We optimize the net surface-atmosphere CO flux, rather than just fire CO. Posterior fluxes are then decomposed into fossil fuel, fire, and Biogenic sources using the fractional contribution to each gridcell. This is a commonly used approach within the CO inversion community. Please see the following text in the revised text:

Sec 2: "For each inversion, the combined carbon emissions released as both CO and CO₂ (CO₂+CO) are then estimated using the CO₂/CO emission factors from the same bottom-up database. The CO₂/CO emission ratios can be highly variable, adding uncertainty to our analysis. We incorporate some of this uncertainty here as each bottom-up database has different mean emission ratios for Canadian forests (range: 7.7--10.8 gC of CO₂ per gC of CO)."

We have also elaborated on this in Sec. 7.4: *"We estimate posterior CO₂ fluxes from the posterior CO emissions using the CO₂/CO emission ratios provided by the prior GFED4.1s, GFAS, and QFED inventories. Each inventory has different CO₂/CO emission; thus, we use the emission ratio to estimate the posterior CO₂ from the same inventory that was used as the prior inventory. This incorporates some uncertainty CO₂/CO emission ratio into the CO₂ emission estimates."*

Supplementary information

Is there any reason why the author did not provide information about the validation of the posterior CO fields against the assimilated MOPITT XCO retrievals, and TCCON?

The MOPITT XCO inversions were not performed for 2023, there were only TROPOMI inversions. Please see the revised supplement, where we give statistics for the data-model mismatch for each of the new TROPOMI inversions.

In academic writing, it would be better to avoid using the phrase "too low" and instead opt for a more precise and formal expression. You could rephrase the sentence to something like:

"The prior inventories indicate significant positive biases, suggesting that the model-simulated XCO levels are underestimated."

Done.

Specific comments on the text:

In Section 2, paragraph 1, Line 5, the authors says that bottom-up inventories vary significantly in global and regional trace gas and aerosol emission estimates. I would add that the top-down approaches also vary significantly.

It is correct that global-scale fire emissions can vary significantly between top-down studies due to factors like the prescription of the OH sink. However, for isolated sources like the Canadian wildfires, our results show that top-down estimates are more precise while including Bayesian and systematic error sources.

Throughout the text, I found that Supplementary Figures are not well referenced. For example, in this section of the text: 'Western Quebec (49°–55° N, 72°–80° W), which is typically relatively wet (Fig. S3),' the authors reference Fig. S3, but this Figure has 3 panels. It would be clearer for the reader if they specified which panel they're referring to. I'm assuming they mean Fig. S3a?

We have specified the panel and improved references throughout.

References:

Palmer, T. E., McSweeney, C. F., Booth, B. B. B., Priestley, M. D. K., Davini, P., Brunner, L., Borchert, L., and Menary, M. B.: Performance-based sub-selection of CMIP6 models for impact assessments in Europe, *Earth Syst. Dynam.*, 14, 457–483, <https://doi.org/10.5194/esd-14-457-2023>, 2023.

Peiro, H., Crowell, S., and Moore III, B.: Optimizing 4 years of CO₂ biospheric fluxes from OCO-2 and in situ data in TM5: fire emissions from GFED and inferred from MOPITT CO data, *Atmos. Chem. Phys.*, 22, 15817–15849, <https://doi.org/10.5194/acp-22-15817-2022>, 2022.

Reviewer Reports on the First Revision:

Referees' comments:

Referee #1 (Remarks to the Author):

I am satisfied that the authors have adequately updated the calculation and presentation regarding the five key points raised in my review of the initial submission.

Here a few minor editorial recommendations:

(Page and line numbers refer to the version with tracked changes.)

p.3, 3rd-last line: delete "supplementary"

p. 8, l. 8: "differences in" -> "differences between" and be explicit which two quantities differ.

p. 10: "Z-scores are calculated from the median of" appear to contradict Eq. (3), which uses the mean. Please check and be consistent on median vs. mean.

p. 12 "we calculate observational errors that incorporate errors in the atmospheric transport model": Uncertainty estimates are used in flux inversions to balance the information used from model and observations. Mixing the the error calculations is a basic contradiction to this underlying principles of flux inversions. As a result, this will give unjustified weight to information from the model. However, the presented study shows that the results are still mostly constrained by the observations. Therefore, this apparent error in the error characterisation should not have a significant effect on the results and does not need to be corrected. But a clarifying sentences might help readers who want to build their own work on the presented methodology.

p. 17, l.4: "inversion" -> "inversions"

Fig. S1: If the authors have fixed the error in their inversions with GFAS in the meantime, an update of this figure in the final manuscript would be nice.

Referee #2 (Remarks to the Author):

Thanks for your detailed response to the original reviews.

I would argue that our Canadian forests are already more likely to be a carbon source as opposed to a carbon sink due to fire, insects and drought (Kurz et al 2008a; Kurz et al. 2008b). This is the reason Canada opted out of Kyoto article 3.4. Thus, the question about our Canadians forests being a long-term carbon sink has already been answered.

Jain et al. (2024 preprint) have an updated preliminary 2023 area burned that is lower than what this paper cites. Also, there is updated evacuees' information in the preprint.

The idea that emissions from can be of global significance has been addressed in papers addressing peat fires (Page et al. 2002)

What role might pyrocumulonimbus (pyrocbs) play in emissions estimates. Some of the busiest days of the fire season were dominated by pyrocbs (record breaking year – see Jain et al. 2024) and the emissions from fires would be in thunderstorms (pyrocbs).

Minor points

Out of curiosity, why not use ERA5 reanalysis instead of MERRA-2

References

Jain et al. 2024. Canada Under Fire – Drivers and Impacts of the Record-Breaking 2023 Wildfire Season. Preprint. 95878fe74ae00b234311dbe287cf9613.pdf (d197for5662m48.cloudfront.net)

Kurz et al. 2008a. Mountain pine beetle and forest carbon feedback to climate change. *Nature* 452: 987-990

Kurz et al. 2008b. Risk of natural disturbances makes future contribution of Canada's forests to the global carbon cycle highly uncertain. *PNAS* 105: 1551-1555

Page SE, Siegert F, Rieley JO, Boehm HD, Jaya A, Limin S. The amount of carbon released from peat and forest fires in Indonesia during 1997. *Nature*. 2002 Nov 7;420(6911):61-5. doi: 10.1038/nature01131. PMID: 12422213.

Referee #3 (Remarks to the Author):

I have reviewed the revised version of your manuscript titled "Unprecedented Canadian Forest Fire Carbon Emissions During 2013" by Byrne et al. (2024), and the authors have satisfactorily addressed all the questions and concerns raised in the previous round of review.

The additional sensitivity experiments related to the sources of errors in the inversion and the improvements to the method section add robustness to the results and conclusions drawn in the manuscript.

I have only a few comments on Section 5, Figure 4. Please ensure that the units reported are correct. While the Y-axis indicates that the units are Tg C, I notice that the grey line indicates "total CO₂ emissions." Is the grey line reported in Tg C? Canada's official NGHG inventory reports their emissions in Mt CO₂eq. The authors should clearly state the units used for their estimates and how these units can be translated for policy discussions. This is particularly relevant and important for the public audience and government inventory compilers, who often get confused by what is reported by atmospheric inversions. It would be helpful to add something to the text like: "All quantities presented are in units of teragrams of carbon (Tg C, 10⁹ g C), so units of megatonnes of CO₂ (or 1 million tonnes of CO₂) used in policy are equal to 3.664 multiplied by the value in units of Tg C."

Additionally, note that Figure 2 contains a typographical error: "Fig??"

Apart from the comments above, I appreciate the effort all authors have put into addressing each point raised, and I am satisfied with the changes made.

I have no further comments or suggestions for improvement.

Author Rebuttals to First Revision:

We thank the referees for their thoughtful comments. We have addressed the issues brought up by the referees in our revised manuscript, with point-by-point responses below. Reviewer comments are in black and our responses in blue.

Referees' comments:

Referee #1 (Remarks to the Author):

I am satisfied that the authors have adequately updated the calculation and presentation regarding the five key points raised in my review of the initial submission.

Here a few minor editorial recommendations:

(Page and line numbers refer to the version with tracked changes.)

p.3, 3rd-last line: delete "supplementary"

done

p. 8, l. 8: "differences in" -> "differences between" and be explicit which two quantities differ.

Replaced

"This difference in categorization leads to large differences in the Canadian carbon budget."

With

"This difference in categorization leads to large differences between the Canadian NGHGI and an estimate using the IPCC guideline definitions"

p. 10: "Z-scores are calculated from the median of" appear to contradict Eq. (3), which uses the mean. Please check and be consistent on median vs. mean.

This sentence has been further clarified:

"Z-scores are calculated from the median of the CMIP6 ensemble by calculating the temporal mean of the median model over 2000-2019 for a given grid cell"

p. 12 "we calculate observational errors that incorporate errors in the atmospheric transport model": Uncertainty estimates are used in flux inversions to balance the information used from model and observations. Mixing the the error calculations is a basic contradiction to this underlying principles of flux inversions. As a result, this will give unjustified weight to information from the model. However, the presented study shows that the results are still mostly constrained by the observations. Therefore, this apparent error in the error characterisation should not have a significant effect on the results and does not need to be corrected. But a clarifying sentences might help readers who want to build their own work on the presented methodology.

The inversions construction employed in this study follows the standard implementation. The cost function includes two terms: one term quantifies the perturbation to the prior normalized by the uncertainties in the prior, and the other term measures the differences between observations and simulated concentrations by transport model normalized by the observational error. This latter term includes both errors in observations and forward model, which is the atmospheric transport model error in this context. The incorporation of forward model errors into the observational error is discussed in more detail in Sec. 3.1.1 of Rodgers (2000).

Rodgers, Clive D. *Inverse methods for atmospheric sounding: theory and practice*. Vol. 2. World scientific, 2000.

p. 17, l.4: "inversion" -> "inversions"

fixed

Fig. S1: If the authors have fixed the error in their inversions with GFAS in the meantime, an update of this figure in the final manuscript would be nice.

Unfortunately, this was not re-run. Given resource constraints, we concluded that re-running this experiment did not significantly add to the study

Referee #2 (Remarks to the Author):

Thanks for your detailed response to the original reviews.

I would argue that our Canadian forests are already more likely to be a carbon source as opposed to a carbon sink due to fire, insects and drought (Kurz et al 2008a; Kurz et al. 2008b). This is the reason Canada opted out of Kyoto article 3.4. Thus, the question about our Canadians forests being a long-term carbon sink has already been answered.

We have acknowledged this assessment in the revised manuscript:

“And while Canadian forests have historically experienced large stand replacing fires at infrequent intervals of 30 to more than 100 years [32–34], increases in fire frequency will likely reduce biomass recovery and impact species composition [34–37]. It has also been argued that fire, insects, and drought may already be driving Canadian forests into a carbon source [38, 39]. In the extreme case that expansive fires.”

[38] Kurz, W.A., Dymond, C., Stinson, G., Rampley, G., Neilson, E., Carroll, A., Ebata, T., Safranyik, L.: Mountain pine beetle and forest carbon feedback to climate change. *Nature* 452(7190), 987–990 (2008) <https://doi.org/10.1038/nature0677735>

[39] Kurz, W.A., Stinson, G., Rampley, G.J., Dymond, C.C., Neilson, E.T.: Risk of natural disturbances makes future contribution of Canada’s forests to the global carbon cycle highly uncertain. *Proceedings of the National Academy of Sciences* 105(5), 1551–1555 (2008) <https://doi.org/10.1073/pnas.070813310>

Jain et al. (2024 preprint) have an updated preliminary 2023 area burned that is lower than what this paper cites. Also, there is updated evacuees’ information in the preprint.

Thank you, we have update this information.

The idea that emissions from can be of global significance has been addressed in papers addressing peat fires (Page et al. 2002)

The sentence “This comparison demonstrates that fire events of this magnitude are relevant to the global carbon budget.” was redundant and removed.

What role might pyrocumulonimbus (pyrocbs) play in emissions estimates. Some of the busiest days of the fire season were dominated by pyrocbs (record breaking year – see Jain et al. 2024) and the emissions from fires would be in thunderstorms (pyrocbs).

Pyrocumulonimbus cause rapid vertical motions that are not captured by the CMS-Flux system. To assess the impact of this strong convection on observed CO we performed model runs where the emissions were released at a plume injection height simulated by IS4FIRES (see supplementary tables). Although this sensitivity test showed some difference in the observation-model mismatch, these were quite small relative to differences between prior and posterior.

Minor points

Out of curiosity, why not use ERA5 reanalysis instead of MERRA-2

This was an arbitrary decision. One motivation is that MERRA-2 reanalysis is used to perform transport in the CMS-Flux system, so this provides the same set of meteorological fields for two applications in the study.

References

Jain et al. 2024. Canada Under Fire – Drivers and Impacts of the Record-Breaking 2023 Wildfire Season. Preprint. 95878fe74ae00b234311dbe287cf9613.pdf (d197for5662m48.cloudfront.net)

Kurz et al. 2008a. Mountain pine beetle and forest carbon feedback to climate change. *Nature* 452: 987-990

Kurz et al. 2008b. Risk of natural disturbances makes future contribution of Canada's forests to the global carbon cycle highly uncertain. *PNAS* 105: 1551-1555

Page SE, Siegert F, Rieley JO, Boehm HD, Jaya A, Limin S. The amount of carbon released from peat and forest fires in Indonesia during 1997. *Nature*. 2002 Nov 7;420(6911):61-5. doi: 10.1038/nature01131. PMID: 12422213.

Referee #3 (Remarks to the Author):

I have reviewed the revised version of your manuscript titled "Unprecedented Canadian Forest Fire Carbon Emissions During 2013" by Byrne et al. (2024), and the authors have satisfactorily addressed all the questions and concerns raised in the previous round of review.

The additional sensitivity experiments related to the sources of errors in the inversion and the improvements to the method section add robustness to the results and conclusions drawn in the manuscript.

I have only a few comments on Section 5, Figure 4. Please ensure that the units reported are correct. While the Y-axis indicates that the units are Tg C, I notice that the grey line indicates "total CO₂ emissions." Is the grey line reported in Tg C? Canada's official NGHGI reports their emissions in Mt CO₂eq. The authors should clearly state the units used for their estimates and how these units can be translated for policy discussions. This is particularly relevant and important for the public audience and government inventory compilers, who often get confused by what is reported by atmospheric inversions. It would be helpful to add something to the text like: "All quantities presented are in units of teragrams of carbon (Tg C, 10⁹ g C), so units of megatonnes of CO₂ (or 1 million tonnes of CO₂) used in policy are equal to 3.664 multiplied by the value in units of Tg C."

We have confirmed the plotted units are correct. We have also revised the caption to make this point:

"Canada's NGHGI CO₂ emissions and removals compared with the 2023 Canadian Fires. Lines show the annual net emissions or removals from managed forest land (green), harvested wood products (brown), natural disturbances that are not counted towards Canada's emissions (red), and the economy-wide net CO₂ emissions in grey. The top-down estimates of the 2023 CO₂+CO fires emissions over managed land are shown in black. Total CO₂ emissions, HWP, and forest land emissions and removals were obtained from Table A11-1 of the NGHGI \cite{ECCC-2023}, while natural disturbances were obtained from table 6-5. Note that all quantities presented are in units of teragrams of carbon (1 TgC = 1 MtC = 10¹² gC), which can be converted to units of megatonnes of CO₂ (MtCO₂) by multiplying by a factor of 3.664."

Additionally, note that Figure 2 contains a typographical error: "Fig??"

Fixed.

Apart from the comments above, I appreciate the effort all authors have put into addressing each point raised, and I am satisfied with the changes made.

I have no further comments or suggestions for improvement.